# Uncertainty Matters in Dynamic Gaussian Splatting for Monocular 4D Reconstruction

**Fengzhi Guo**[1]  **Chih-Chuan Hsu**[1]  **Sihao Ding**[2]  **Cheng Zhang**[1]

[1]Texas A&M University  [2]Mercedes-Benz Research & Development North America

## Abstract

Reconstructing dynamic 3D scenes from monocular input is fundamentally under-constrained, with ambiguities arising from occlusion and extreme novel views. While dynamic Gaussian Splatting offers an efficient representation, vanilla models optimize all Gaussian primitives uniformly, ignoring whether they are well or poorly observed. This limitation leads to motion drifts under occlusion and degraded synthesis when extrapolating to unseen views. We argue that uncertainty matters: Gaussians with recurring observations across views and time act as reliable anchors to guide motion, whereas those with limited visibility are treated as less reliable. To this end, we introduce USPLAT4D, a novel **U**ncertainty-aware dynamic Gaussian **Splat**ting framework that propagates reliable motion cues to enhance **4D** reconstruction. Our approach estimates time-varying per-Gaussian uncertainty and leverages it to construct a spatio-temporal graph for uncertainty-aware optimization. Experiments on diverse real and synthetic datasets show that explicitly modeling uncertainty consistently improves dynamic Gaussian Splatting models, yielding more stable geometry under occlusion and high-quality synthesis at extreme viewpoints. Project page: https://tamu-visual-ai.github.io/usplat4d/.

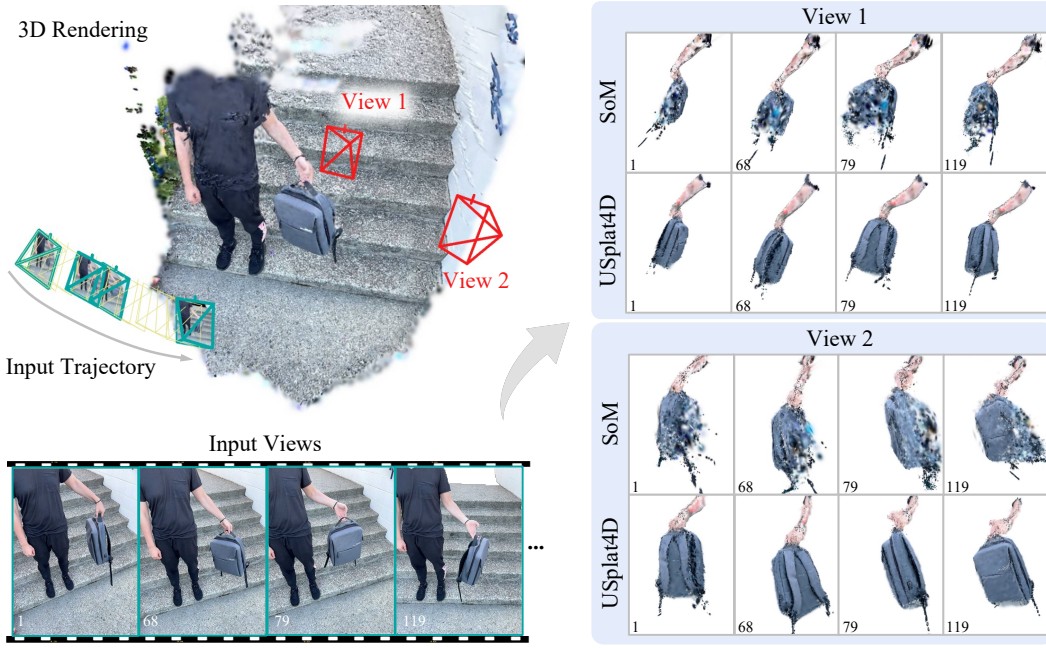

Figure 1: We show a challenging case from the DyCheck dataset (Gao et al., 2022), where a person casually rotates a backpack while being captured by a moving monocular camera. The goal is to reconstruct the dynamic object for arbitrary viewpoints and timestamps. The state-of-the-art dynamic Gaussian Splatting methods, e.g., Shape-of-Motion (SoM) (Wang et al., 2025a), struggle on extreme novel views far from the input trajectory, such as opposite-side views (view 1) or large angle offsets (view 2). We propose USPLAT4D, an **U**ncertainty-aware dynamic Gaussian **Splat**ting model that produces more accurate and consistent **4D** reconstruction. **Left**: rendered dynamic scene from our model for illustration alongside four sampled RGB inputs. **Right**: novel view synthesis at two extreme novel viewpoints. Please refer to the project page for clearer visual comparison.

# 1 INTRODUCTION

Reconstructing dynamic 3D scenes from monocular input is a fundamental problem across a variety of tasks, including augmented reality, robotics, and human motion analysis (Slavcheva et al., 2017; Li et al., 2023; Newcombe et al., 2015; Joo et al., 2014; Gao et al., 2022). However, despite its wide applicability, monocular dynamic reconstruction remains highly challenging (Liang et al., 2025c), particularly under occlusion and extreme viewpoint changes. Recently, the advent of 3D Gaussian Splatting (Kerbl et al., 2023) has enabled real-time photorealistic rendering and sparked a series of dynamic extensions (Luiten et al., 2024; Lei et al., 2025; Stearns et al., 2024; Liu et al., 2025; Duan et al., 2024; Huang et al., 2024; Yang et al., 2024b;c; Li et al., 2024; Sun et al., 2024b; Duisterhof et al., 2023; Das et al., 2024; Lin et al., 2024; Wu et al., 2025). These methods parameterize motion with shared canonical fields (Wu et al., 2024; Yang et al., 2024c; Liang et al., 2025b; Guo et al., 2024; Lu et al., 2024; Wan et al., 2024; Liu et al., 2025), deformation bases (Wang et al., 2025a; Das et al., 2024; Lin et al., 2024; Li et al., 2024), or direct 4D modeling (Duan et al., 2024; Yang et al., 2024b).

Despite their differences in formulation, existing dynamic Gaussian splatting methods often share a common assumption: motion is optimized uniformly across all Gaussians using 2D supervision such as depth (Yang et al., 2024a), optical flow (Teed & Deng, 2020), and photometric consistency (Doersch et al., 2023). This uniform treatment overlooks that some Gaussians are strongly constrained by recurring observations, while others are only weakly constrained. As a result, motion estimates drift under occlusion and synthesized views degrade at novel viewpoints.

To maintain spatio-temporal consistency, we argue that confidently observed Gaussians should be prioritized and used to guide the optimization of less reliable ones. Consider the example in Figure 1, where a rotating backpack is captured by a moving monocular camera. At any moment, a portion of the surface is self-occluded and invisible. Yet, humans can readily infer their appearance and motion by recalling previously observed surfaces and extrapolating with temporal continuity. Such ability anchors on the most reliable parts of the backpack, i.e., those clearly observed from other viewpoints and timestamps. This suggests a key principle: when observations are partial, reconstruction should be guided by confident cues and propagated structurally to uncertain regions.

Building on this insight, we propose USPLAT4D, a novel **U**ncertainty-aware dynamic Gaussian **Splat**ting framework for monocular **4D** reconstruction. We first introduce a principled method to estimate time-varying uncertainty for each Gaussian, capturing how reliably it is constrained by recurring observations. This uncertainty then guides the selection of anchor Gaussians and propagate motion across space and time. To realize this principle, we organize Gaussians into a spatio-temporal graph, where uncertainty determines node importance, edge construction, and adaptive loss weighting. The goal of the uncertainty-aware graph optimization is to ensure that confident parts of the scene dynamically guide the reconstruction of the rest, even in occluded or unseen views.

We validate our approach on various real and synthetic datasets on monocular 4D reconstruction. We show that explicitly leveraging uncertainty significantly enhances both motion tracking and novel view synthesis, with particularly strong gains under extreme viewpoints. Our framework, including uncertainty estimation, graph construction, and adaptive training, is model-agnostic and can be integrated into existing dynamic Gaussian splatting pipelines that parameterize per-Gaussian motion. Overall, USPLAT4D introduces a principled way to model uncertainty in dynamic Gaussian Splatting, yielding more stable motion estimates under occlusion and high-quality extreme view synthesis.

# 2 RELATED WORK

**Dynamic Gaussian splatting.** Recent advances in dynamic Gaussian Splatting have enabled *monocular* 4D reconstruction via per-Gaussian deformation or canonical motion modeling (Liang et al., 2025c; Yang et al., 2024b;c; Wu et al., 2024; Li et al., 2024) or high-fidelity dynamic scene reconstruction from *multi-view* inputs (Luiten et al., 2024; Wang et al., 2025b). To reconstruct 4D GS from the monocular video, methods such as SoM (Wang et al., 2025a), MoSca (Lei et al., 2025), Marbles (Stearns et al., 2024), and 4D-Rotor (Duan et al., 2024) use low-rank motion bases to regularize deformation, while others model canonical flows (Liang et al., 2025b; Liu et al., 2025). Although they demonstrate high-fidelity rendering on near-input validation views, they do not explicitly model the motion behind occluders or identify reliable Gaussians for motion guidance. MoSca (Lei et al., 2025) introduces a soft motion score but lacks structured propagation. In contrast, USPLAT4D

selects high-confidence Gaussians and constructs an uncertainty-aware motion graph to propagate motion through spatio-temporally coherent connections, improving robustness in occluded regions and enabling localized refinement beyond low-rank modeling (Kim et al., 2024; Huang et al., 2024).

**Uncertainty estimation in scene reconstruction.** Uncertainty modeling has been widely explored in neural rendering (Shen et al., 2021; Li et al., 2022; Pan et al., 2022; Shen et al., 2022; Kim et al., 2022; Zhan et al., 2022; Yan et al., 2023; Lee et al., 2022; Sünderhauf et al., 2023), particularly to improve robustness under occlusion, sparse views, and ambiguity. To avoid overfitting on reconstructing the static scene, SE-GS (Zhao et al., 2025) designs an uncertainty-aware perturbing strategy by estimating the self-ensembling uncertainty. In dynamic settings, uncertainty has been used to smooth motion or reweight gradients (Kim et al., 2024), but typically as an auxiliary signal decoupled from the underlying motion representation. In contrast, USPLAT4D treats uncertainty as a central modeling component. We estimate confidence of each Gaussian and use it to guide key node selection, edge construction, and loss weighting in a spatio-temporal graph. This allows high-confidence Gaussians to guide motion propagation while reducing the influence of uncertain regions. To the best of our knowledge, this is among the first attempts and analyses to model the uncertainty and directly integrate it into graph-based motion modeling for dynamic reconstruction.

## 3 PRELIMINARY

We first review dynamic Gaussian Splatting and its learning objective to establish the notation for Section 4. Our method is model-agnostic, building on this standard formulation and applicable to a wide range of dynamic Gaussian Splatting variants (Lei et al., 2025; Wang et al., 2025a).

**Dynamic 3D Gaussians.** Vanilla dynamic Gaussian Splatting (Luiten et al., 2024) represents a scene with a set of time-varying 3D Gaussians. Formally, the state of a Gaussian at time $t$ is defined as

$$\mathbf{G}_t = (\mathbf{p}_t, \mathbf{q}_t, \mathbf{s}, \alpha, \mathbf{c}), \tag{1}$$

where $\mathbf{p}_t \in \mathbb{R}^3$ denotes the position at time $t$, $\mathbf{q}_t \in \mathbb{R}^4$ the quaternion rotation, $\mathbf{s} \in \mathbb{R}^3$ the scale, $\alpha \in \mathbb{R}$ the opacity, and $\mathbf{c} \in \mathbb{R}^{N_c}$ the color coefficients (e.g., spherical harmonics or RGB), $N_c$ the color dimension. The trajectory of a Gaussian is then given by the sequence $\{\mathbf{G}_t\}_{t=1}^{T}$, where $T$ is the number of frames. While early extensions introduce time-varying color (Yang et al., 2024b), this hinders 3D motion tracking. Recent methods, such as SoM (Wang et al., 2025a) and MoSca (Lei et al., 2025), restrict the color space to stabilize motion estimation and enable reliable tracking.

**Learning objectives.** To optimize the 4D Gaussian field, existing methods minimize a combination of losses. A photometric reconstruction loss enforces consistency between rendered and ground-truth images, while motion-locality losses regularize the temporal evolution of Gaussians. These locality terms include isometry, rigidity, relative rotation, velocity, and acceleration constraints (Lei et al., 2025; Huang et al., 2024), which shrink the large motion search space and stabilize optimization.

**Limitations.** Although effective near input views, these objectives remain fragile under occlusion and extreme novel viewpoints, as they rely heavily on unstable 2D priors such as depth, optical flow, or photometric consistency. As a result, reconstructions often drift over time and lose geometric consistency across different views. To overcome this challenge, we introduce a dynamic uncertainty model that explicitly encodes the reliability of each Gaussian over time and forms the foundation of our framework. In particular, Gaussians are partitioned into key and non-key nodes and connected through an uncertainty-weighted graph, which enforces spatio–temporal consistency.

## 4 UNCERTAINTY-AWARE DYNAMIC GAUSSIAN SPLATTING

**Overview.** Given a monocular video, we begin by building on vanilla dynamic Gaussian Splatting models to estimate a time-varying uncertainty score for each Gaussian, explicitly capturing its reliability across frames (Section 4.1). These uncertainty scores then guide the construction of an uncertainty-weighted graph that systematically organizes Gaussians into key and non-key nodes (Section 4.2). The resulting graph subsequently drives an optimization process that propagates reliable motion cues to uncertain regions, thereby refining both motion estimation and rendering quality of the dynamic scene (Section 4.3). An overview of the entire pipeline is illustrated in Figure 2.

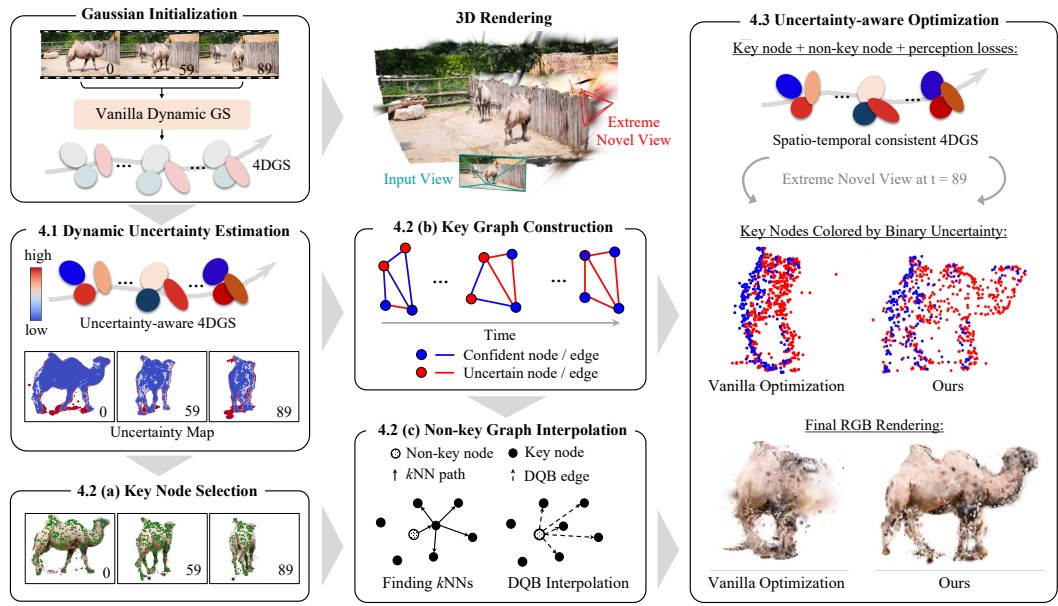

Figure 2: **Overview of the proposed** USPLAT4D. We first estimate time-varying uncertainty for each Gaussian (Section 4.1). We then leverage these uncertainties to select reliable Gaussians as key nodes, while others are treated as non-key nodes for graph construction (Section 4.2). Finally, we optimize the spatio-temporal graph with uncertainty-weighted losses, yielding consistent 4D Gaussians (Section 4.3). The right column shows that our approach significantly improves novel view renderings compared to vanilla optimization.

## 4.1 DYNAMIC UNCERTAINTY ESTIMATION

Vanilla dynamic Gaussian Splatting optimizes all primitives uniformly, even though some are well observed across time while others remain ambiguous. This causes drift under occlusion and instability at extreme viewpoints. We therefore assign each Gaussian $\mathbf{G}_i$ a time-varying uncertainty to estimate its reliability per frame and use it to guide optimization in a model-agnostic way.

**Per-Gaussian scalar uncertainty.** A straightforward way to capture reliability is to assign each Gaussian $i$ a scalar uncertainty $u_{i,t}$ at every frame $t$. Intuitively, if a primitive is frequently and clearly observed, its uncertainty should be small; if it is rarely visible or weakly constrained, its uncertainty should be large. The photometric loss is defined as:

$$\mathcal{L}_{2,t} = \sum_{h \in \Omega} \|\bar{C}_t^h - C_t^h\|_2^2, \quad \text{where} \quad C_t^h = \sum_{i=1}^{N_g} T_{i,t}^h \alpha_i \, c_i. \tag{2}$$

Here, $\Omega$ is the pixel index set, $\bar{C}_t^h$ and $C_t^h$ denote the ground-truth and rendered colors of pixel $h$ at frame $t$, and $c_i$ is the color parameter of Gaussian $i$. The rendered pixel color is obtained by $\alpha$-blending, where the blending weight is given by $T_{i,t}^h \alpha_i$, with $T_{i,t}^h$ the transmittance of Gaussian $i$ at pixel $h$, $\alpha_i$ its opacity, and $N_g$ is the number of Gaussians. By differentiating $\mathcal{L}_{2,t}$ with respect to $c_i$ and applying the local minimum assumption, we obtain the closed-form variance estimate (please see Appendix A.1 for detailed derivation):

$$\sigma_{i,t}^2 = \left( \sum_{h \in \Omega_{i,t}} (T_{i,t}^h \alpha_i)^2 \right)^{-1}, \tag{3}$$

where $\Omega_{i,t} \subseteq \Omega_t$ is the set of pixels contributed to by that Gaussian. We thus take this variance as the scalar uncertainty, i.e., $u_{i,t} := \sigma_{i,t}^2$. However, the local minimum assumption may not hold everywhere. To account for unconverged pixels, we introduce an indicator function to test per-pixel convergence:

$$\mathbb{1}_t(h) = \begin{cases} 1 & \text{if } \|\bar{C}_t^h - C_t^h\|_1 < \eta_c, \\ 0 & \text{otherwise,} \end{cases} \tag{4}$$

where $\eta_c > 0$ is a color-error threshold. For Gaussian $i$ at frame $t$, we define the aggregate indicator $\mathbb{1}_{i,t} = \prod_{h \in \Omega_{i,t}} \mathbb{1}_t(h)$, which equals 1 only if all covered pixels are convergent. If $\mathbb{1}_{i,t} = 0$, we assign a large constant $\phi$ to indicate high uncertainty. Therefore, the final scalar uncertainty is:

$$u_{i,t} = \mathbb{1}_{i,t}\,\sigma_{i,t}^2 + (1 - \mathbb{1}_{i,t})\,\phi. \tag{5}$$

This design jointly adjusts the strength according to convergence status: Gaussians that are well supported by observations receive low $u_{i,t}$, reflecting high reliability, while unreliable ones are assigned high values, which enables trustworthy primitives to guide ambiguous ones during optimization.

**From scalar to depth-aware uncertainty.** While the scalar definition is intuitive, it implicitly assumes that uncertainty is isotropic in 3D space. This is problematic in monocular settings, where depth is much less reliable than image-plane coordinates. A uniform scalar tends to be over-confident along the camera axis, leading to geometric distortion. To address this, we propagate image-space errors into 3D and represent each Gaussian by an anisotropic uncertainty matrix:

$$\mathbf{U}_{i,t} = \mathbf{R}_{wc}\,\mathbf{U}_c\,\mathbf{R}_{wc}^{\mathsf{T}}, \quad \text{where} \quad \mathbf{U}_c = \mathrm{diag}(r_x u_{i,t},\, r_y u_{i,t},\, r_z u_{i,t}). \tag{6}$$

Here, $\mathbf{R}_{wc}$ is the camera-to-world rotation and $r_x, r_y, r_z$ are axis-aligned scaling factors. Note that only rotation is required to propagate uncertainty, since translation does not affect covariance. This transforms 2D uncertainty into axis-aligned 3D uncertainty, incorporating both the camera pose and the directional sensitivity of depth. A typical example is the "Camel" sequence in Figure 2 (also see the supplementary video): without depth-aware uncertainty, the camel's body shrinks unnaturally, whereas our formulation preserves its correct shape.

## 4.2 UNCERTAINTY-ENCODED GRAPH CONSTRUCTION

Per-Gaussian uncertainty in Equation 6 provides a local measure of reliability, but treating Gaussians independently cannot guarantee spatio–temporal consistency. Neighboring primitives often share correlated motion, and reliable ones should anchor the optimization of uncertain ones. Prior graph-based methods (Huang et al., 2024; Lei et al., 2025) attempt to capture this correlation, e.g., MoSca (Lei et al., 2025) introduces a 3D lifting graph. In contrast, we build the graph directly on uncertainty: Gaussians are ranked by reliability and partitioned into key and non-key nodes, so that stable primitives drive motion propagation while ambiguous ones are regularized. To realize this, we design an uncertainty-aware graph that encodes reliability in both node selection and edge connectivity, providing the foundation for the optimization described in the following sections.

**Graph definition.** We represent the scene with a directed graph $\mathcal{G} = (\mathcal{V}, \mathcal{E})$, where each node $i \in \mathcal{V}$ corresponds to a Gaussian $\mathbf{G}_i$ and edges $(i,j) \in \mathcal{E}$ encode spatial affinity and motion similarity. Crucially, nodes are partitioned into a small key set $\mathcal{V}_k$ and a large non-key set $\mathcal{V}_n$ according to their uncertainties $\{u_{i,t}\}$ from Section 4.1. We define key nodes as stable Gaussians that carry strong motion cues across time and views, while non-key nodes inherit motion from their key neighbors.

**Key node selection.** Our key node selection operates in 3D space. We employ the following two-stage strategy that balances spatial coverage and temporal stability:

1. *Candidate sampling via 3D gridization.* At each frame, we partition the scene into a 3D voxel grid. Voxels containing only high-uncertain Gaussians[1] are discarded. In the remaining grids, which may contain multiple low-uncertainty Gaussians, we randomly select one per grid. This per-voxel "uniform selection" ensures spatial coverage and reduces redundancy, unlike random selection without voxel uniformity[2]. This is based on the assumption that each meaningful motion will occupy a unique voxel in at least one frame; motions indistinguishable at this resolution are minor and handled by non-key interpolation (see below). Intuitively, distinct motions leave separable footprints, while small differences are smoothed out.
2. *Thresholding by significant period.* For each Gaussian candidate, we compute its *significant period*, defined as the number of frames where its uncertainty stays below a threshold. We retain

---

[1]We maintain a typical 1:49 key/non-key ratio by selecting the top 2% (around 1000-th) most confident Gaussians. Ablations with ratios from 0.5%~4% (please see Appendix D.1) show consistent performance across different scenes and models, with 2% lying on a stable plateau balancing coverage and reliability.

[2]We test random selection with the same per-frame count but without per-grid uniformity. This produces non-uniform spatial distributions (i.e., some voxels oversampled, others missed). See Table 3(d) for analysis.

only those with a significant period of at least 5 frames, ensuring that key nodes have sufficient temporal support to contribute reliably to motion estimation. Candidates with insufficient temporal coverage tend to destabilize graph optimization and yield under-constrained solutions.

**Edge construction.** We construct edges separately for *key* and *non-key* nodes, as their roles differ: key nodes provide structural anchors for motion propagation, while non-key nodes interpolate appearance locally. The key graph captures long-range geometric and motion dependencies (e.g., both ends of a limb moving coherently), which existing distance-based heuristics such as local $k$NN (Huang et al., 2024) or global min–max distances (Lei et al., 2025) cannot robustly handle.

To address this, we adopt an Uncertainty-Aware $k$NN (UA-$k$NN). For a key node $i$, we select neighbors only among other low-uncertainty key nodes, evaluated at its most reliable frame $\hat{t} = \arg\min_t\{u_{i,t}\}$, and measure distances with uncertainty weighting to favor trustworthy connections:

$$\mathcal{E}_i = k\mathrm{NN}_{j\in\mathcal{V}_k\setminus\{i\}}\Big(\|\mathbf{P}_{i,\hat{t}} - \mathbf{P}_{j,\hat{t}}\|_{(\mathbf{U}_{w,\hat{t},i}+\mathbf{U}_{w,\hat{t},j})}\Big). \tag{7}$$

Here, the Mahalanobis metric up-weights directions of high uncertainty, so edges are formed between nodes that are both spatially close and reliable. As will be shown in Section 4.3, these edges are further pruned by the key graph loss for additional robustness and to prevent spurious long-range connections. For a non-key node $i$, we assign it to its closest key node across the sequence:

$$j = \arg\min_{l\in\mathcal{V}_k}\sum_{t=0}^{T-1}\|\mathbf{P}_{i,t} - \mathbf{P}_{l,t}\|_{(\mathbf{U}_{w,t,i}+\mathbf{U}_{w,t,l})} \tag{8}$$

and connect $\mathcal{E}_i = \mathcal{E}_j \cup \{j\}$, where $j$ is the closest key nodes. Intuitively, each uncertain non-key node is attached to the most reliable key node that stays close to it over time, so its motion can be regularized by stable anchors for consistency. In both cases, uncertainty-aware $k$NN ensures edges are anchored by reliable nodes, promoting stable motion propagation and preventing drift from uncertain regions.

## 4.3 UNCERTAINTY-AWARE OPTIMIZATION

Vanilla optimization (Section 3) of dynamic Gaussians often fails under occlusion or extreme viewpoints. This leads unreliable primitives to drift, since they are optimized as strongly as reliable ones. To address this, we incorporate uncertainty into the objective: key nodes with stable observations serve as anchors, while non-key nodes are regularized more softly through interpolation. We design separate objectives for key and non-key nodes, then unify them in a total loss.

**Key node loss.** Key nodes are low-uncertainty Gaussians that anchor motion. We encourage them to stay close to their pre-optimized positions:

$$\mathcal{L}^{\mathrm{key}} = \sum_{t=0}^{T-1}\sum_{i\in\mathcal{V}_k}\|\mathbf{P}_{i,t} - \mathbf{p}_{i,t}^{\mathrm{o}}\|_{\mathbf{U}_{w,t,i}^{-1}} + \mathcal{L}^{\mathrm{motion,key}}, \tag{9}$$

where $\mathbf{U}_{w,t,i}$ down-weights directions of high uncertainty, ensuring motion is corrected mainly along reliable axes. Superscript o denotes the parameters from the pretrained Gaussian Splatting model, which serves as initialization before our uncertainty-aware optimization. The key node motion loss $\mathcal{L}^{\mathrm{motion,key}}$ regularizes the temporal evolution of Gaussians by isometry, rigidity, rotation, velocity, and acceleration constraints, which is discussed in the Appendix A.2 in detail.

**Non-key node loss.** Non-key nodes are interpolated from nearby key nodes using Dual Quaternion Blending (DQB) (Kavan et al., 2007), which provides smooth motion by blending their neighbors:

$$\Big(\mathbf{p}_{i,t}^{\mathrm{DQB}}, \mathbf{q}_{i,t}^{\mathrm{DQB}}\Big) = \mathrm{DQB}\Big(\{(w_{ij}, \mathbf{T}_{j,t})\}_{j\in\mathcal{E}_i}\Big), \tag{10}$$

where $w_{ij}$ are normalized edge weights for blending, $\mathbf{T}_{j,t} \in \mathbb{SE}(3)$ is the transform of key node $j$, and $\mathbf{p}_{i,t}^{\mathrm{DQB}}$ and $\mathbf{q}_{i,t}^{\mathrm{DQB}}$ are the position and rotation of Gaussian $i$ at time $t$ acquired by DQB, respectively. We then regularize non-key nodes to both their initialization and interpolated trajectory:

$$\mathcal{L}^{\mathrm{non\text{-}key}} = \sum_{t=0}^{T-1}\sum_{i\in\mathcal{V}_n}\|\mathbf{P}_{i,t} - \mathbf{p}_{i,t}^{\mathrm{o}}\|_{\mathbf{U}_{w,i}^{-1}} + \sum_{t=0}^{T-1}\sum_{i\in\mathcal{V}_n}\|\mathbf{P}_{i,t} - \mathbf{p}_{i,t}^{\mathrm{DQB}}\|_{\mathbf{U}_{w,i}^{-1}} + \mathcal{L}^{\mathrm{motion,non\text{-}key}}. \tag{11}$$

Here, $\mathcal{L}^{\mathrm{motion,non\text{-}key}}$ is the non-key node motion loss, which will also be discussed in Appendix A.2. This non-key loss keeps non-key nodes close to their pretrained states while aligning them with motions propagated from reliable key nodes, preventing drift while ensuring coherence.

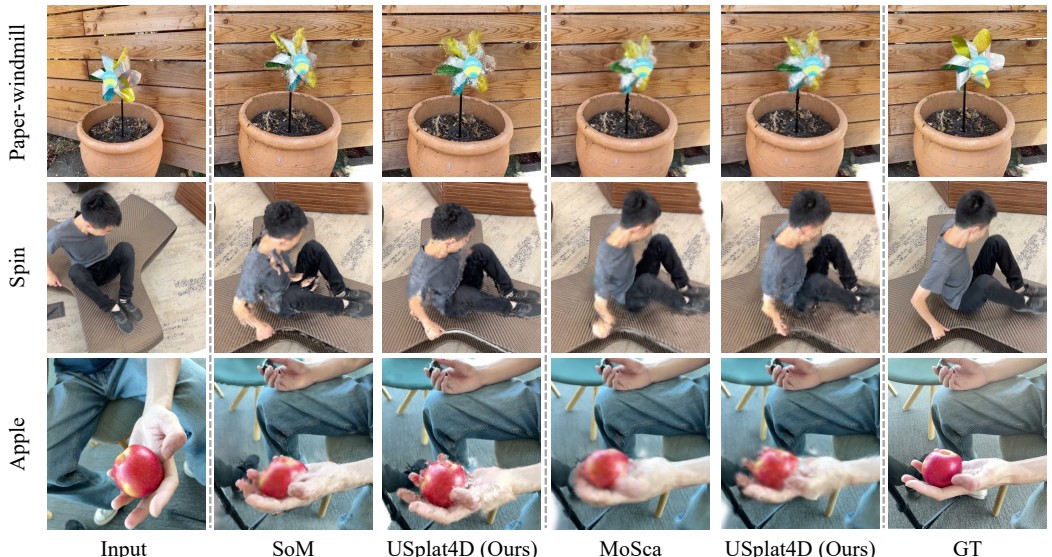

Figure 3: **Qualitative results on validation views of the DyCheck dataset** (Gao et al., 2022). We show comparisons with two strong baselines, SoM (Wang et al., 2025a) and MoSca (Lei et al., 2025). USPLAT4D improves visual quality and better preserves geometry (e.g., arms in "Spin", hands in "Space-out") and pose (e.g., "Windmill"). Please zoom in for details. See supplementary for more examples and other baselines.

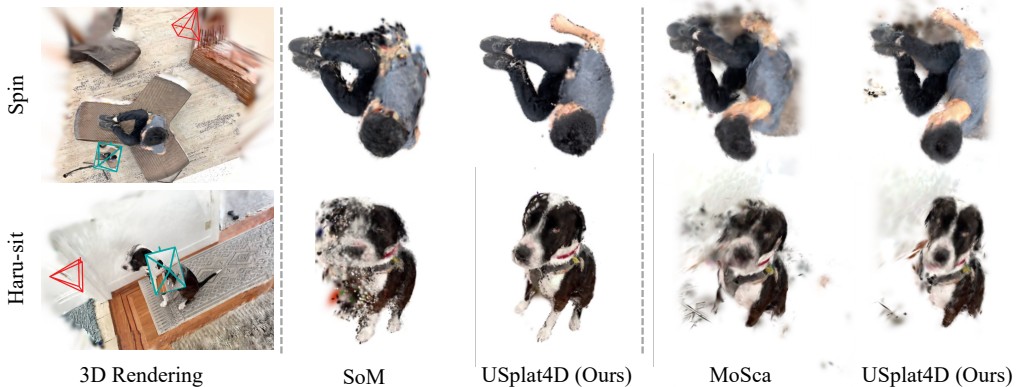

Figure 4: **Qualitative results on extreme novel views from DyCheck.** Unlike Figure 3, these manually sampled extreme views (red cameras) lack ground-truth. USPLAT4D better preserves fine structures (e.g., the dog's head in "haru-sit") and occluded regions (e.g., the hand in "spin") under large viewpoint shifts.

**Total loss.** The final objective combines the key and non-key node losses with the photometric loss:

$$\mathcal{L}^{\text{total}} = \mathcal{L}^{\text{rgb}} + \mathcal{L}^{\text{key}} + \mathcal{L}^{\text{non-key}}. \tag{12}$$

Overall, the uncertainty in our framework serves three purposes: (1) re-weighting deviations of key nodes, (2) guiding the interpolation of non-key nodes, and (3) balancing their influence in the total loss. Our design mitigates drift under occlusion and maintains geometric consistency at novel views.

## 5 EXPERIMENTS

### 5.1 SETUP

**Datasets. (1)** DyCheck (Gao et al., 2022): We follow prior works (Wang et al., 2025a; Lei et al., 2025; Stearns et al., 2024) to evaluate on 7 scenes with validation views. Since these validation views are near the input views, we additionally sample extreme novel views for qualitative analysis.

Table 1: **Quantitative results on DyCheck**. We report results on 5 scenes at 1× resolution and 7 scenes at 2× resolution, following existing protocols. USPLAT4D consistently outperforms state-of-the-art Gaussian Splatting based methods. See Figure 3 for qualitative results on validation views and Figure 4 for extreme views.

| Setting | Method | mPSNR↑ | mSSIM↑ | mLPIPS↓ |
|---|---|---|---|---|
| 5 scenes 1 × resolution | SC-GS (Huang et al., 2024) | 14.13 | 0.477 | 0.49 |
| | Deformable 3DGS (Yang et al., 2024c) | 11.92 | 0.490 | 0.66 |
| | 4DGS (Wu et al., 2024) | 13.42 | 0.490 | 0.56 |
| | MoDec-GS (Kwak et al., 2025) | 15.01 | 0.493 | 0.44 |
| | MoBlender (Zhang et al., 2025) | 16.79 | **0.650** | **0.37** |
| | SoM (Wang et al., 2025a) | 16.72 | 0.630 | 0.45 |
| | USPLAT4D (ours) | **16.85** | **0.650** | 0.38 |
| 7 scenes 2 × resolution | Dynamic Gaussians (Luiten et al., 2024) | 7.29 | – | 0.69 |
| | 4DGS (Wu et al., 2024) | 13.64 | – | 0.43 |
| | Gaussian Marbles (Stearns et al., 2024) | 16.72 | – | 0.41 |
| | MoSca (Lei et al., 2025) | 19.32 | 0.706 | 0.26 |
| | USPLAT4D (ours) | **19.63** | **0.716** | **0.25** |

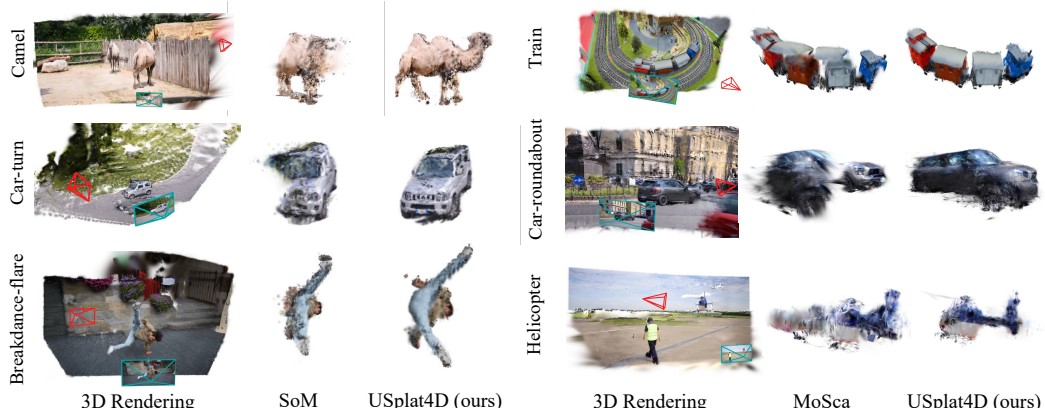

Figure 5: **Qualitative results on extreme novel views from DAVIS**. For each case, we show an input-view rendering and compare the baseline (SoM (Wang et al., 2025a) or MoSca (Lei et al., 2025)) with our USPLAT4D on an extreme novel view (red). USPLAT4D yields clearer reconstructions under challenging conditions.

**(2)** DAVIS (Perazzi et al., 2016): To test generalization across different scenarios, we qualitatively evaluate on challenging monocular videos using DAVIS dataset, which cover non-rigid motion, occlusions, and complex dynamics. **(3)** Objaverse (Deitke et al., 2023): We construct a synthetic benchmark[3] from Objaverse by selecting 6 challenging articulated objects with diverse textures and motions. Specifically, we set cameras to follow a circular trajectory of 121 frames (3° per step) around each object, always facing toward it. For validation, we render views at horizontal angular offsets of 30° increments from 30° to 330°, each with a fixed elevation of 35°, in order to capture the object's 3D structure from diverse viewpoints. Please see Appendix C.4 for additional details.

**Baselines.** We evaluate USPLAT4D against state-of-the-art dynamic Gaussian Splatting methods. We use SoM (Wang et al., 2025a) and MoSca (Lei et al., 2025) as our main base models. Specifically, SoM is a widely adopted standard in the field, and MoSca represents the current state of the art. Notably, our framework is agnostic to the underlying architecture and is compatible with any method that estimates per-Gaussian motion. Please see Appendix B for additional implementation details (e.g., hyper-parameters for model training) and details of additional baseline methods.

---

[3]Existing benchmarks lack ground-truth for extreme novel views, making systematic evaluation difficult. We therefore build a synthetic benchmark, where strong results support our generalization claims (see experiments). Note that such synthetic setups are widely used to stress-test models (Yao et al., 2025; Liang et al., 2024).

Table 2: **Results on the Objaverse dataset.** We evaluate novel view synthesis across increasing horizontal angular ranges: $(0°, 60°]$, $(60°, 120°]$, and $(120°, 180°]$. USPLAT4D consistently improves over SoM (Wang et al., 2025a) and MoSca (Lei et al., 2025), with gains most pronounced at larger viewpoint shifts.

| Method | View Range $(0°, 60°]$ | | | View Range $(60°, 120°]$ | | | View Range $(120°, 180°]$ | | |
|---|---|---|---|---|---|---|---|---|---|
| | PSNR↑ | SSIM↑ | LPIPS↓ | PSNR↑ | SSIM↑ | LPIPS↓ | PSNR↑ | SSIM↑ | LPIPS↓ |
| SoM | 16.09 | 0.860 | 0.31 | 15.58 | 0.854 | 0.32 | 16.45 | 0.858 | 0.31 |
| USPLAT4D | **16.63**$_{.55}$ | **0.866**$_{.007}$ | **0.27**$_{.03}$ | **16.57**$_{.09}$ | **0.868**$_{.014}$ | **0.27**$_{.05}$ | **17.03**$_{.58}$ | **0.872**$_{.014}$ | **0.26**$_{.05}$ |
| MoSca | 16.18 | 0.881 | 0.24 | 15.74 | 0.875 | 0.25 | 15.89 | 0.876 | 0.25 |
| USPLAT4D | **16.22**$_{.04}$ | **0.885**$_{.004}$ | **0.22**$_{.02}$ | **15.98**$_{.24}$ | **0.884**$_{.009}$ | **0.23**$_{.02}$ | **16.31**$_{.42}$ | **0.886**$_{.011}$ | **0.21**$_{.03}$ |

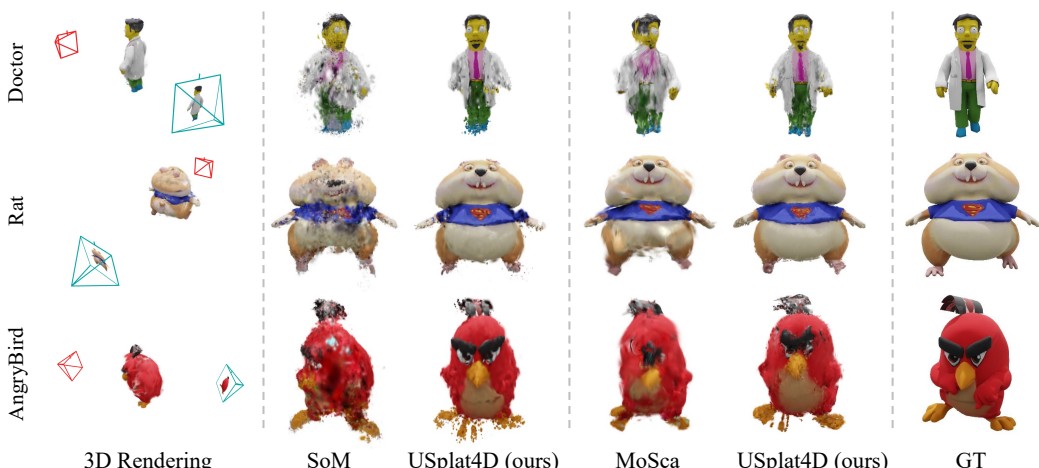

Figure 6: **Qualitative results on Objaverse.** Each case shows a 3D rendering from an input view and a comparison between the baseline (SoM (Wang et al., 2025a) or MoSca (Lei et al., 2025)) and our USPLAT4D at an extreme novel view (red). Please see the supplementary video for clearer visualization.

## 5.2 MAIN RESULTS

**Results on DyCheck.** Table 1 reports results on DyCheck validation views, which are close to the input trajectories. USPLAT4D consistently outperforms baselines across all metrics; corresponding qualitative examples are shown in Figure 3. However, these validation views remain relatively easy due to the proximity to the input views. The more significant improvements emerge under extreme novel viewpoints (Figure 4), which are not included in Table 1 but are essential for assessing robustness under severe viewpoint shifts. Our results avoids the collapse and distortions observed in the baselines. Please refer to Appendix C for the tracking results and per-scene comparison.

**Qualitative results of extreme novel view synthesis on DAVIS.** We evaluate on selected monocular videos from the DAVIS dataset, which include fast motion, deformation, and self-occlusion. As ground-truth geometry is unavailable, we focus on qualitative comparisons. As shown in Figure 5, USPLAT4D yields more plausible geometry and coherent reconstructions under extreme viewpoint shifts, where the baseline often exhibits distortion, blur, or artifacts.

**Results on Objaverse.** Table 2 shows that USPLAT4D consistently surpasses SoM (Wang et al., 2025a) and MoSca (Lei et al., 2025), with gains most pronounced under large viewpoint shifts (120°–180°). Figure 6 confirms these improvements qualitatively: our method preserves geometry and textures under extreme novel views, where baselines often blur or collapse. Please refer to Appendix C for more quantitative and qualitative results.

**Other results.** We provide further results and analysis in Appendix C, including per-scene results on the DyCheck and Objaverse, evaluations on the NVIDIA (Yoon et al., 2020) and HyperNeRF (Park et al., 2021) datasets, results of the tracking task, and comparisons with additional baseline methods.

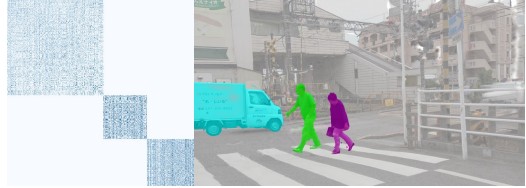

Figure 7: **Key node weight matrix and Gaussian segmentation.** The weight matrix visualizes edge connections among key nodes. Nodes belonging to different diagonal blocks do not share weights, indicating the no connections across those blocks. The images further illustrate the corresponding Gaussian segmentation, where colors represent the block assignment of each key and non-key node.

## 5.3 ABLATION AND ANALYSIS

We conduct ablation studies to assess key design choices in USPLAT4D, using MoSca (Lei et al., 2025) as the base model and evaluating on the DyCheck validation set (Gao et al., 2022). As described in Section 4, our core design include the graph initialization strategy and the use of uncertainty cues during graph construction and optimization. Finally, we analyze the model's capabilities.

**Ablation study on uncertainty usage and key node selection.** Table 3 shows that uncertainty and key node selection are essential across key components of our framework. **(a)** When removed from key node selection and replaced with uniform 2D sampling, the graph fails to emphasize well-constrained Gaussians, leading to unstable anchors and degraded propagation. **(b)** Replacing UA-$k$NN with distance-only $k$NN weakens graph connectivity, as edges disregard node reliability and mistakenly connect unstable primitives.

Table 3: **Ablation study on uncertainty usage and key node selection.** We assess the impact of uncertainty estimation and key node selection strategy by removing them from key components in USPLAT4D individually.

| Ablation Setting | PSNR↑ | SSIM↑ | LPIPS↓ |
|---|---|---|---|
| USPLAT4D (full model) | **19.63** | **0.716** | **0.25** |
| (a) w/o key node uncertainty | 18.86 | 0.688 | 0.28 |
| (b) w/o UA-$k$NN | 19.50 | 0.711 | 0.26 |
| (c) w/o loss weighting | 19.08 | 0.681 | 0.25 |
| (d) w/o 3D gridization | 19.50 | 0.712 | 0.25 |

**(c)** Excluding uncertainty weighting from the training loss (i.e., applying the DQB loss without $\mathbf{U}$) reduces PSNR/SSIM, since unreliable Gaussians are updated as aggressively as reliable ones, causing drift across frames. **(d)** Replacing the key node candidate sampling strategy from 3D grid-based sampling with spatially random sampling leads to performance degradation, since the non-uniform spatial distribution neither guarantees sufficient spatial coverage nor avoids redundant samples. Please see Appendix D for additional ablation study (on hyperparameter key / non-key ratio, color threshold, significant period threshold, and scaling factor) and discussion on time complexity.

**Further analysis.** Our model naturally supports multiple objects and inherently group Gaussians by motion. In key and non-key graphs, nodes with similar instance-level motion are strongly connected, while nodes from different instances have weak (or few) connections. As shown in Figure 7, the key-graph weight matrix (defined in Appendix A.2.1) becomes approximately block-diagonal after reordering nodes with standard community detection (e.g., spectral clustering (Von Luxburg, 2007)). After clustering key nodes, we assign non-key nodes to the same motion groups using their non-key graph connections. Each Gaussian gets a label, and when objects move differently, the segmentation closely matches instance segmentation with dynamic tracking. Besides, we further discuss the challenging cases including textureless region, fast motion, and deforming objects in Appendix E.

## 6 DISCUSSION AND CONCLUSION

We introduce USPLAT4D, a novel dynamic Gaussian Splatting framework that demonstrates that how *uncertainty estimation* is pivotal for monocular 4D reconstruction. By integrating time-varying uncertainty estimation with an uncertainty-guided graph, our approach significantly improves geometric consistency and rendering quality, particularly under challenging extreme novel view synthesis. While our framework remains influenced by the computational overhead and inherent errors of underlying visual foundation models, we hope that our findings could encourage further work on leveraging uncertainty to advance robust 4D reconstruction and extreme novel view synthesis.

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

# APPENDIX

## A   ADDITIONAL DETAILS ON USPLAT4D

### A.1   DERIVATION OF UNCERTAINTY (SECTION 4.1 OF THE MAIN PAPER)

To derive our uncertainty model, we begin by analyzing the color blending function used in volumetric rendering. To simplify the notation, we omit time $t$. The rendered pixel color $C$ along a ray is computed as a weighted sum of the color contributions from all Gaussians:

$$C^h = \sum_{i=1}^{N_g} T_i^h \alpha_i c_i := \sum_{i=1}^{N_g} v_i^h c_i, \tag{S1}$$

where $i$ is the index of Gaussians, $T_i$ is the accumulated transmittance up to the $i$-th Gaussian, $\alpha_i$ is the opacity, $c_i$ is the color, and $v_i := T_i \alpha_i$ denotes the blending weight. To estimate the color uncertainty, we derive a closed-form expression for each Gaussian's color via maximum likelihood estimation (MLE) under an RGB $\ell_2$ loss:

$$\mathcal{L}_2 = \sum_{h \in \Omega} \|\bar{C}^h - C^h\|_2^2 \tag{S2}$$

Here, $h \in \Omega$ indexes the set of all pixels, $\bar{C}^h$ is the ground-truth color, and $C^h$ is the color rendered using the blending model in Eq. S1.

To find the optimal color $c_k$ for the $k$-th Gaussian, we compute the gradient of the loss with respect to $c_k$ and obtain:

$$\begin{aligned} \left.\frac{\partial \mathcal{L}(\mathcal{D}; \boldsymbol{\theta})}{\partial c_k}\right|_{\boldsymbol{\theta}_{\text{MLE}}} &= -\sum_{h \in \Omega} 2(\bar{C}^h - C^h) v_k^h \\ &= -\sum_{h \in \Omega} 2(b_k^h - v_k^h c_k) v_k^h, \end{aligned} \tag{S3}$$

where $\mathcal{D}$ is the training dataset, $\boldsymbol{\theta}$ is the learnable model parameter vector, and we define $b_k^h := \bar{C}^h - \sum_{j \neq k} v_j^h c_j$ to isolate the contribution of the $k$-th Gaussian to the rendered color at pixel $h$. Assuming a local minimum, the MLE estimate satisfies:

$$\left.\frac{\partial \mathcal{L}(\mathcal{D}; \boldsymbol{\theta})}{\partial c_k}\right|_{\theta_{\text{MLE}}} = 0 \tag{S4}$$

Solving this closed-form yields the optimal color for Gaussian $k$:

$$c_k = \frac{\sum_{h \in \Omega} b_k^h v_k^h}{\sum_{h \in \Omega} (v_k^h)^2} \tag{S5}$$

Finally, by assuming $b_k^h \sim \mathcal{N}(v_k^h \bar{c}_k, \epsilon^2)$ where $\bar{c}_k$ is the groundtruth Gaussian color and $\text{Var}(b_k^h) = \epsilon^2$, the corresponding closed-form Gaussian color uncertainty (i.e., $\sigma_k^2 := \text{Var}(c_k | v_k^h)$) under the unit-variance Gaussian noise model (i.e., $\epsilon^2 = 1$) can be derived as:

$$\sigma_k^2 = \left(\sum_{h \in \Omega} (v_k^h)^2\right)^{-1} \tag{S6}$$

### A.2   DETAILS OF TRAINING LOSSES (SECTION 3 AND SECTION 4.3 OF THE MAIN PAPER)

As described in Section 4.3 and Eq. 12 of the main paper, our total loss is:

$$\mathcal{L}^{\text{total}} = \mathcal{L}^{\text{rgb}} + \mathcal{L}^{\text{key}} + \mathcal{L}^{\text{non-key}} \tag{S7}$$

where $\mathcal{L}^{\text{rgb}}$ is the perception loss and $\mathcal{L}^{\text{motion,key}}$ and $\mathcal{L}^{\text{motion,non-key}}$ are the motion loss used in $\mathcal{L}^{\text{key}}$ and $\mathcal{L}^{\text{non-key}}$, which covers the motion locality of key and non-key nodes, respectively. In this section, we provide additional details.

### A.2.1 MOTION LOSS

The vanilla Gaussian splatting approach (e.g., (Luiten et al., 2024)) represents a dynamic scene using 3D Gaussians with time-varying motion and define motion loss. The recent paper also follow up on the motion loss and modifies it based on their tasks. Here, we give the general form of these motion losses. The isometry loss describing the distance constraint between Gaussians is defined as

$$\mathcal{L}^{\text{iso}} = \frac{1}{k|\mathcal{V}|} \sum_{t=0}^{T-1} \sum_{i \in \mathcal{V}} \sum_{j \in \text{knn}_{i;k}} w_{i,j}(\|\mathbf{p}_{j,o} - \mathbf{p}_{i,o}\|_2 - \|\mathbf{p}_{j,t} - \mathbf{p}_{i,t}\|_2) \quad \text{(S8)}$$

where subscript $o$ means canonical space, $\mathbf{t}$ is the Gaussian's 3D position vector, and $w_{i,j}$ represent the edge weight between $i$-th and $j$-th Gaussians. Since isometry does not take coordinates into consideration, we use rigidity loss to unify the coordinates and constrain the relative motion between Gaussians. The rigidity loss is defined by

$$\mathcal{L}^{\text{rigid},\Delta} = \frac{1}{k|\mathcal{V}|} \sum_{t=\Delta}^{\mathbf{T}-1} \sum_{i \in \mathcal{V}} \sum_{j \in \text{knn}_{i;k}} w_{i,j} \left\| \left( \mathbf{p}_{j,t-\Delta} - \mathbf{T}_{i,t-\Delta} \mathbf{T}_{i,t}^{-1} \mathbf{p}_{j,t} \right) \right\|_2 \quad \text{(S9)}$$

where $\mathbf{T}_{i,t}$ is the $i$-th Gaussian transformation matrix at $t$-th time index and $\Delta$ is the time interval. With a larger weight $w_{i,j}$, the Gaussian pair is more rigid. Beyond that, we also constrain the relative rotation explicitly for finer control on the rotation penalty. The relative rotation loss is defined by

$$\mathcal{L}^{\text{rot},\Delta} = \frac{1}{k|\mathcal{V}|} \sum_{t=\Delta}^{T-1} \sum_{i \in \mathcal{V}} \sum_{j \in \text{knn}_{i;k}} w_{i,j} \left\| \mathbf{q}_{j,t} \mathbf{q}_{j,t-\Delta}^{-1} - \mathbf{q}_{i,t} \mathbf{q}_{i,t-\Delta}^{-1} \right\|_2 \quad \text{(S10)}$$

where $\mathbf{q}$ is the quaternion representation of the rotation matrix. All there three loss define the deformation of the Gaussians. Besides, the object motion in the world space tends to be smooth, so we define the velocity and acceleration to regularize the Gaussian model to avoid overfitting on the training views. The velocity loss is defined as

$$\mathcal{L}^{\text{vel}} = \sum_{t=1}^{T-1} \sum_{i \in \mathcal{V}} \|\mathbf{p}_{i,t-\Delta} - \mathbf{p}_{i,t}\|_1 + \|\mathbf{q}_{i,t-1} \mathbf{q}_{i,t}^{-1}\|_1 \quad \text{(S11)}$$

The acceleration loss is defined as

$$\mathcal{L}^{\text{acc}} = \sum_{t=2}^{T-1} \sum_{i \in \mathcal{V}} \|\mathbf{p}_{i,t-2} - 2\mathbf{p}_{i,t-1} + \mathbf{p}_{i,t}\|_1 + \|\mathbf{q}_{i,t-2} \mathbf{q}_{i,t-1}^{-1} (\mathbf{q}_{i,t-1} \mathbf{q}_{i,t}^{-1})^{-1}\|_1 \quad \text{(S12)}$$

The motion loss used in the recent works (Lei et al., 2025; Stearns et al., 2024; Huang et al., 2024; Luiten et al., 2024) can be obtained by the different combinations of these losses. Then, the motion locality loss is defined as

$$\mathcal{L}^{\text{motion}} = \lambda^{\text{iso}} \mathcal{L}^{\text{iso}} + \lambda^{\text{rigid}} \mathcal{L}^{\text{rigid}} + \lambda^{\text{rot}} \mathcal{L}^{\text{rot}} + \lambda^{\text{vel}} \mathcal{L}^{\text{vel}} + \lambda^{\text{acc}} \mathcal{L}^{\text{acc}}, \quad \text{(S13)}$$

where $\lambda$ is the hyperparameter. Specifically, we set $\lambda^{\text{iso}}$ and $\lambda^{\text{rigid}}$ to 1 to ensure the geometry preservation and assign $\lambda^{\text{rot}}$, $\lambda^{\text{vel}}$, and $\lambda^{\text{acc}}$ to 0.01 for rigid orientation and motion smoothness.

### A.2.2 PERCEPTION LOSS

The perception loss $\mathcal{L}^{\text{rgb}}$ includes a standard combination of RGB $\ell_1$ loss and SSIM loss. Following the base models (SoM and MoSca), we also incorporate 2D prior losses, i.e., mask loss, depth loss, depth gradient loss, and tracking loss, into $\mathcal{L}^{\text{rgb}}$. We emphasize that these additional terms are not part of our proposed method, but are standard components inherited from the respective base models to ensure consistent training behavior.

## B IMPLEMENTATION DETAILS

USPLAT4D is compatible with dynamic Gaussian splatting methods that provide initial motion parameters. In this section, we describe implementation details for integrating USPLAT4D with two strong base models (i.e., SoM (Wang et al., 2025a) and MoSca (Lei et al., 2025)) along with our training and evaluation protocols.

### B.1 ADAPTATION OF SOM AND MOSCA

Both SoM (Wang et al., 2025a) and MoSca (Lei et al., 2025) adopt low-rank motion parameterizations to achieve compact, smooth, and rigid 4D representations. We first convert the outputs of SoM or MoSca into the data structure required by USPLAT4D. Specifically, we extract the 4D Gaussian primitives $\{\mathbf{G}_i\}_{i=1}^{N_g}$ from each method and reformat them into a unified representation that supports our graph construction and uncertainty modeling pipeline. To mitigate inconsistencies in Gaussian scale and distribution, we unify the spatial volume when selecting key Gaussians and normalize the uncertainty threshold used in both key node selection and edge construction. This ensures that our method behaves consistently across different base models. Since our method introduces additional optimization on top of the pretrained SoM and MoSca models, we ensure fair comparison by continuing to train the original SoM and MoSca with the *same number of additional iterations as our baselines.*

### B.2 TRAINING USPLAT4D MODEL

During preprocessing of USPLAT4D, we sequentially initialize the key and non-key graphs. To train the USPLAT4D model, we build on pretrained base models and allocate additional training iterations for fair comparison. Specifically, we train SoM for 400 extra epochs and MoSca for 1600 extra steps, using a consistent batch size of 8. These schedules are empirically chosen to ensure good convergence while aligning with the respective optimization routines of the base models. For the first 10% and the last 20% of the training duration, we disable both density control and opacity reset to maintain stability. For the remaining iterations, we enable density control and opacity reset to improve rendering quality. The per-epoch and per-step training time is similar to that of SoM and MoSca, respectively.

## C ADDITIONAL EXPERIMENTAL RESULTS

### C.1 COMPARISON WITH ADDITIONAL NERF-BASED METHODS ON DYCHECK DATASET

In Table 1 of the main paper, we primarily compare with Gaussian splatting based methods that use explicit 3D representations. Here, we extend the comparison by including additional NeRF-based methods (Miao et al., 2024; Park et al., 2021; Gao et al., 2022; Li et al., 2023; Sun et al., 2024a; Kappel et al., 2025; Zhao et al., 2024) that adopt implicit neural radiance fields. While these methods are effective for static or mildly dynamic scenes, our experiments show that Gaussian splatting based approaches consistently outperform NeRF-based methods on the DyCheck dataset (Gao et al., 2022). This comparison serves as an extension of Table 1 and the results are summarized in Table S1.

### C.2 TRACKING RESULTS ON THE DYCHECK DATASET

We report 3D keypoint tracking results in Table S2, following the evaluation protocols of MoSca and SoM. Our method achieves consistent improvements in all tracking metrics, including Percentage of Correct Keypoints (PCK) @ (5%, 5cm, 10cm) and End-Point Error (EPE). When applied to MoSca (Lei et al., 2025), our method yields higher PCK@5% (+2.1). Integrated with SoM (Wang et al., 2025a), USPLAT4D achieves notable gains in EPE and PCK at both 5cm and 10cm thresholds, especially improving PCK@5cm by over 11.4%. These results show that our uncertainty-aware graph construction not only enhances visual quality but also improves spatio-temporal consistency for dynamic object tracking.

### C.3 PER-SCENE RESULTS ON THE DYCHECK DATASET

We further present a per-scene breakdown of the DyCheck results reported in Table 1 of the main paper. Results are shown in Table S3. Our method consistently improves per-scene performance across most scenes across PSNR, SSIM, and LPIPS.

Table S1: **Quantitative results on the DyCheck dataset** (Gao et al., 2022). We compare our method applied to two dynamic Gaussian baselines (SoM and MoSca) with NeRF-based and Gaussian-based methods. We report results on 5 scenes at $1\times$ resolution and 7 scenes at $2\times$ resolution, following existing protocols. Our approach consistently improves the base models across all metrics and outperforms other baselines in both settings. Best results are in **bold**, second-best are underlined.

| Setting | Method | mPSNR↑ | mSSIM↑ | mLPIPS↓ |
|---------|--------|--------|--------|---------|
| 5 scenes $1\times$ resolution | DynIBaR (Li et al., 2023) | 13.41 | 0.48 | 0.55 |
| | HyperNeRF (Park et al., 2021) | 15.99 | 0.59 | 0.51 |
| | T-NeRF (Gao et al., 2022) | 15.6 | 0.55 | 0.55 |
| | SC-GS (Huang et al., 2024) | 14.13 | 0.477 | 0.49 |
| | Deformable 3DGS (Yang et al., 2024c) | 11.92 | 0.490 | 0.66 |
| | 4DGS (Wu et al., 2024) | 13.42 | 0.490 | 0.56 |
| | MoDec-GS (Kwak et al., 2025) | 15.01 | 0.493 | 0.44 |
| | MoBlender (Zhang et al., 2025) | 16.79 | **0.650** | **0.37** |
| | HiMoR (Liang et al., 2025a) | – | – | 0.46 |
| | SoM (Wang et al., 2025a) | 16.72 | 0.630 | 0.45 |
| | USPLAT4D (ours) | **16.85** | **0.650** | 0.38 |
| 7 scenes $2\times$ resolution | D-NPC (Kappel et al., 2025) | 16.41 | 0.582 | 0.319 |
| | CTNeRF (Miao et al., 2024) | 17.69 | 0.531 | – |
| | DyBluRF (Sun et al., 2024a) | 17.37 | 0.591 | 0.373 |
| | HyperNeRF (Park et al., 2021) | 16.81 | 0.569 | 0.332 |
| | T-NeRF (Gao et al., 2022) | 16.96 | 0.577 | 0.379 |
| | PGDVS (Zhao et al., 2024) | 15.88 | 0.548 | 0.34 |
| | Dynamic Gaussians (Luiten et al., 2024) | 7.29 | – | 0.69 |
| | 4DGS (Wu et al., 2024) | 13.64 | – | 0.43 |
| | Gaussian Marbles (Stearns et al., 2024) | 16.72 | – | 0.41 |
| | MoSca (Lei et al., 2025) | 19.32 | 0.706 | 0.26 |
| | USPLAT4D (ours) | **19.63** | **0.716** | **0.25** |

Table S2: **Tracking results on the DyCheck dataset** (Gao et al., 2022). We follow the evaluation protocols of MoSca and SoM to report 3D keypoint tracking metrics. USPLAT4D outperforms both baselines by a large margin. Please refer to our supplementary video demo for qualitative results.

| Method | PCK (5%)↑ | Method | EPE↓ | PCK (5cm)↑ | PCK (10cm)↑ |
|--------|-----------|--------|------|-----------|------------|
| MoSca (Lei et al., 2025) | 82.4 | SoM (Wang et al., 2025a) | 0.082 | 43.0 | 73.3 |
| USPLAT4D (ours) | **84.5** | USPLAT4D (ours) | **0.072** | **54.4** | **75.8** |

## C.4 DATA CURATION FOR THE OBJAVERSE DATASET

To evaluate our model under large view shifts, which are rarely present in existing realistic datasets such as DyCheck (Gao et al., 2022) and DAVIS (Perazzi et al., 2016), we curate a high-quality dynamic 3D dataset sourced from Objaverse-1.0 (Deitke et al., 2023). Each curated scene contains a single object with one or more animation labels; by default, we animate the object using its first animation label. As described in Section 5.1 of the main paper, we render 121 frames for each view. Because the total number of animation frames in Objaverse is typically smaller than 121, we cycle through the animation frames in a *zigzag* pattern to maintain motion continuity across the rendered sequence. We place 12 cameras evenly around the object at 30° horizontal intervals and move them along a circular trajectory in a counterclockwise direction with a 3° step per frame. For each frame, we render RGB, depth, and mask images using Blender (Blender Online Community, 2018) and record the corresponding camera intrinsics and extrinsics as ground truth. In our experiments, we use one of the 12 monocular videos as input and utilize the left 11 monocular videos to evaluate rendering quality over three angular ranges: $(0°, 60°]$, $(60°, 120°]$, and $(120°, 180°]$.

Table S3: **Per-scene results on the DyCheck dataset** (Gao et al., 2022). We provide a detailed breakdown of the results summarized in Table 1 of the main paper. ⋆: results reproduced by us, as the original numbers in Table 1 were directly reported from the SoM and MoSca papers. Minor discrepancies may exist due to differences in training. We note that the validation views are near the training views in the DyCheck dataset. Therefore, the observed improvements do not fully reflect the advantages of USPLAT4D in extreme novel view synthesis. Results are formatted as PSNR (↑) / SSIM (↑) / LPIPS (↓).

(a) 7 Scenes, 2 × Resolution

| Method | Apple | Block | Spin | Paper Windmill |
|---|---|---|---|---|
| 4DGS (Wu et al., 2024) | 15.41 / 0.450 / – | 11.28 / 0.633 / – | 14.42 / 0.339 / – | 15.60 / 0.297 / – |
| Gaussian Marbles | 17.70 / 0.492 / – | 17.42 / 0.384 / – | 18.88 / 0.428 / – | 17.04 / 0.394 / – |
| MoSca⋆ (Lei et al., 2025) | 19.46 / 0.809 / 0.34 | 18.17 / 0.678 / 0.32 | 21.26 / 0.752 / 0.19 | 22.36 / 0.743 / 0.16 |
| USPLAT4D (ours) | 19.53 / 0.813 / 0.32 | 18.49 / 0.681 / 0.31 | 21.77 / 0.772 / 0.16 | 22.55 / 0.753 / 0.14 |

| Method | Space-Out | Teddy | Wheel | Mean |
|---|---|---|---|---|
| 4DGS (Wu et al., 2024) | 14.60 / 0.372 / – | 12.36 / 0.466 / – | 11.79 / 0.436 / – | 13.64 / 0.428 / – |
| Gaussian Marbles | 15.94 / 0.435 / – | 13.95 / 0.442 / – | 16.14 / 0.351 / – | 16.72 / 0.418 / – |
| MoSca⋆ (Lei et al., 2025) | 20.50 / 0.664 / 0.26 | 15.54 / 0.626 / 0.35 | 18.16 / 0.684 / 0.23 | 19.35 / 0.706 / 0.26 |
| USPLAT4D (ours) | 20.81 / 0.668 / 0.24 | 15.78 / 0.625 / 0.36 | 18.46 / 0.696 / 0.22 | 19.63 / 0.716 / 0.25 |

(b) 5 Scenes, 1 × Resolution

| Method | Apple | Block | Spin |
|---|---|---|---|
| SC-GS (Huang et al., 2024) | 14.96 / 0.692 / 0.51 | 13.98 / 0.548 / 0.48 | 14.32 / 0.407 / 0.45 |
| MoDec-GS (Kwak et al., 2025) | 16.48 / 0.699 / 0.40 | 15.57 / 0.590 / 0.48 | 15.53 / 0.433 / 0.37 |
| SoM⋆ (Wang et al., 2025a) | 16.56 / 0.749 / 0.54 | 16.27 / 0.652 / 0.43 | 17.32 / 0.710 / 0.29 |
| USPLAT4D (ours) | 16.94 / 0.754 / 0.49 | 16.12 / 0.653 / 0.45 | 17.75 / 0.711 / 0.27 |

| Method | Paper Windmill | Teddy | Mean |
|---|---|---|---|
| SC-GS (Huang et al., 2024) | 14.87 / 0.221 / 0.43 | 12.51 / 0.516 / 0.56 | 14.13 / 0.477 / 0.49 |
| MoDec-GS (Kwak et al., 2025) | 14.92 / 0.220 / 0.38 | 12.56 / 0.521 / 0.60 | 15.01 / 0.493 / 0.44 |
| SoM⋆ (Wang et al., 2025a) | 19.46 / 0.557 / 0.20 | 13.88 / 0.556 / 0.52 | 16.70 / 0.645 / 0.39 |
| USPLAT4D (ours) | 19.69 / 0.555 / 0.19 | 13.77 / 0.551 / 0.50 | 16.85 / 0.645 / 0.38 |

Figure S1: **Additional qualitative results on Objaverse.** Each case shows a 3D rendering from an input view and a comparison between the baseline (SoM (Wang et al., 2025a) or MoSca (Lei et al., 2025)) and our USPLAT4D at an extreme novel view (red). Please see the supplementary video for clearer visualization.

## C.5 ADDITIONAL RESULTS ON THE OBJAVERSE DATASET

Along with the qualitative results in Figure 6 of the main draft, three additional cases are provided in Figure S1 for further comparison. We further provide per-scene results for the Objaverse dataset in Table S4, complementing the summary reported in Table 2 of the main draft. Our method consistently

Table S4: **Per-scene results on the Objaverse dataset.** We provide a detailed breakdown of the results summarized in Table 2 of the main paper. We evaluate novel view synthesis across increasing horizontal angular ranges: $(0°, 60°]$, $(60°, 120°]$, and $(120°, 180°]$.

| Scene | Method | View Range $(0°, 60°]$ | | | View Range $(60°, 120°]$ | | | View Range $(120°, 180°]$ | | |
|---|---|---|---|---|---|---|---|---|---|---|
| | | PSNR↑ | SSIM↑ | LPIPS↓ | PSNR↑ | SSIM↑ | LPIPS↓ | PSNR↑ | SSIM↑ | LPIPS↓ |
| Chick | SoM | 14.26 | **0.852** | 0.34 | 13.86 | 0.845 | 0.37 | 14.70 | 0.847 | 0.36 |
| | USPLAT4D | **14.59** | **0.852** | **0.32** | **14.60** | **0.853** | **0.32** | **15.13** | **0.857** | **0.31** |
| | MoSca | 14.69 | 0.869 | 0.27 | 14.35 | 0.864 | 0.29 | 14.47 | 0.867 | 0.29 |
| | USPLAT4D | **14.82** | **0.880** | **0.24** | **14.42** | **0.876** | **0.27** | **14.97** | **0.878** | **0.25** |
| Panda | SoM | 15.15 | 0.857 | 0.30 | 14.51 | 0.843 | 0.33 | 15.06 | 0.852 | 0.31 |
| | USPLAT4D | **15.51** | **0.873** | **0.26** | **15.28** | **0.874** | **0.26** | **15.38** | **0.874** | **0.26** |
| | MoSca | **15.37** | **0.888** | **0.21** | **14.83** | 0.881 | **0.23** | **15.15** | 0.880 | **0.22** |
| | USPLAT4D | 15.08 | **0.888** | 0.22 | 14.33 | **0.883** | 0.24 | 15.03 | **0.886** | **0.22** |
| Woman | SoM | 18.47 | 0.900 | 0.21 | 17.92 | 0.894 | 0.22 | 19.85 | 0.904 | 0.20 |
| | USPLAT4D | **20.34** | **0.920** | **0.13** | **20.28** | **0.920** | **0.13** | **21.07** | **0.923** | **0.12** |
| | MoSca | 16.67 | **0.910** | 0.15 | 16.48 | 0.908 | 0.15 | 16.79 | 0.910 | 0.14 |
| | USPLAT4D | **17.00** | **0.910** | **0.13** | **17.07** | **0.911** | **0.13** | **17.16** | **0.911** | **0.12** |
| Rat | SoM | 16.61 | **0.833** | 0.36 | 15.81 | **0.837** | 0.38 | 16.08 | **0.835** | 0.37 |
| | USPLAT4D | **16.73** | **0.833** | **0.35** | **16.66** | **0.837** | **0.35** | **17.04** | **0.835** | **0.34** |
| | MoSca | 17.94 | 0.859 | 0.27 | 17.19 | 0.848 | 0.29 | 16.85 | 0.847 | 0.30 |
| | USPLAT4D | **18.53** | **0.873** | **0.23** | **18.09** | **0.872** | **0.24** | **18.27** | **0.874** | **0.24** |
| Doctor | SoM | **17.78** | 0.877 | 0.28 | **17.51** | 0.874 | 0.29 | **18.18** | 0.876 | 0.28 |
| | USPLAT4D | 17.68 | **0.885** | **0.25** | 17.35 | **0.885** | **0.26** | 17.77 | **0.887** | **0.25** |
| | MoSca | **17.12** | 0.896 | 0.23 | 16.16 | 0.887 | 0.26 | 16.76 | 0.888 | 0.24 |
| | USPLAT4D | 17.07 | **0.898** | **0.21** | **16.32** | **0.894** | **0.24** | **17.10** | **0.897** | **0.22** |
| Angry Bird | SoM | 14.23 | 0.839 | 0.34 | 13.85 | 0.831 | 0.34 | 14.84 | 0.836 | 0.32 |
| | USPLAT4D | **15.24** | **0.843** | **0.32** | **15.27** | **0.845** | **0.30** | **15.79** | **0.855** | **0.29** |
| | MoSca | **15.04** | **0.864** | 0.29 | **15.16** | 0.859 | 0.28 | 14.97 | 0.859 | 0.29 |
| | USPLAT4D | 14.78 | **0.864** | **0.28** | 14.60 | **0.862** | **0.28** | **15.07** | **0.868** | **0.26** |

achieves lower LPIPS than both SoM (Wang et al., 2025a) and MoSca (Lei et al., 2025) across all angular ranges and scenes. For PSNR and SSIM, our model is also generally superior to SoM and MoSca in most angular ranges and scenes.

## C.6 RESULTS ON NVIDIA DATASET

Our main quantitative analysis focuses on the DyCheck-iPhone dataset (Gao et al., 2022) in the main paper, which aligns well with our goal of synthesizing dynamic objects captured by a moving camera (see Section 1 of the main paper). Here we also evaluate our method on the NVIDIA dataset (Yoon et al., 2020) for broader comparison. We follow the evaluation pipeline in RoDynRF (Liu et al., 2023). The results are shown in Table S5. We note that USPLAT4D also improves performance when applied to state-of-the-art MoSca (Lei et al., 2025) on the NVIDIA dataset, although the gains are marginal. This is expected, as the NVIDIA dataset (or other datasets such as HyperNeRF (Park et al., 2021) dataset) differs from DyCheck to feature input views with more limited motion. In contrast, datasets with larger camera movement or more challenging dynamics (DyCheck and Objaverse) benefit more from our design, leading to more substantial improvements.

Table S5: **Results on the NVIDIA dataset** (Yoon et al., 2020).

| Method | PSNR↑ | SSIM↑ | LPIPS↓ |
|---|---|---|---|
| MoSca Lei et al. (2025) | 26.77 | 0.854 | **0.07** |
| **MoSca + G2DSplat** | **26.93** | **0.855** | 0.07 |

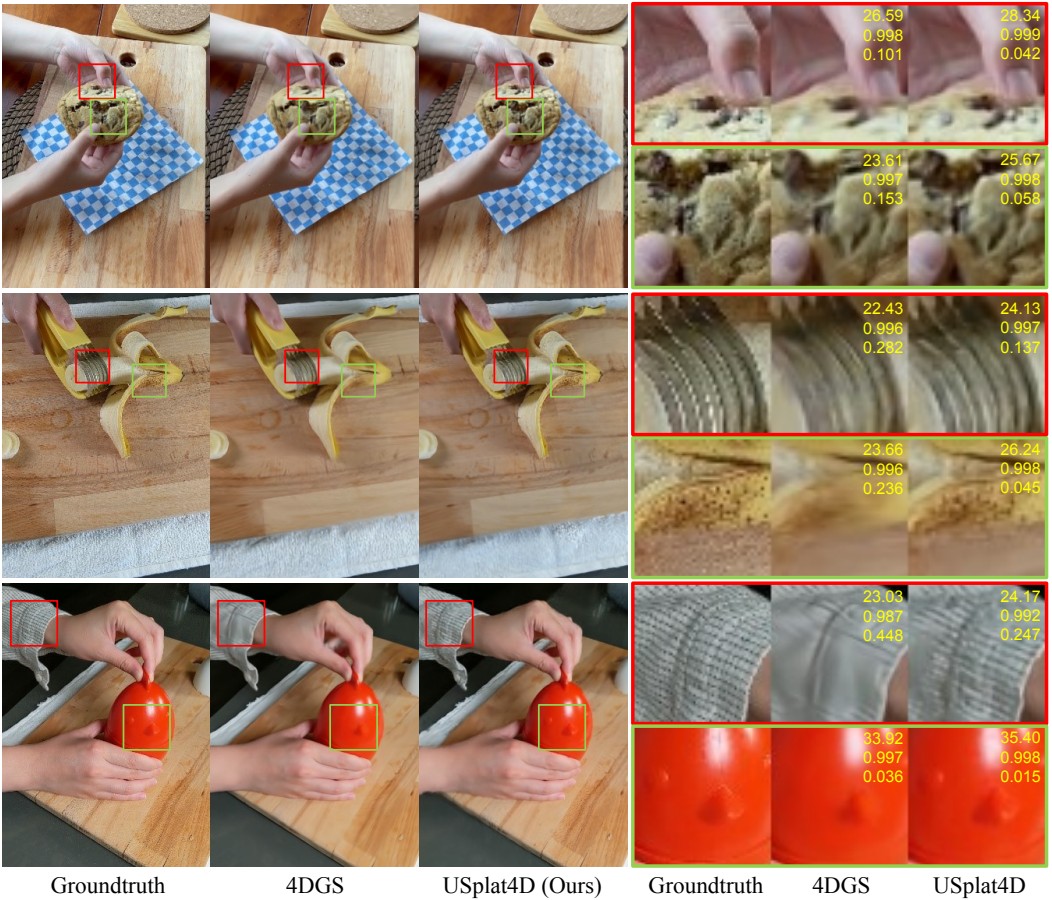

|  Groundtruth | 4DGS | USplat4D (Ours) | Groundtruth | 4DGS | USplat4D |

Figure S2: **Comparison on HyperNeRF dataset.** The groundtruth, 4DGS, and USPLAT4D (ours) are arranged from left to right. The red and green boxes highlights regions with fine-grained details. The values displayed in the top-right corner represent PSNR, SSIM, and LPIPS, arranged from top to bottom. Our method reconstructs high-frequency structures more faithfully and preserves finer geometry compared with both 4DGS.

### C.7 COMPARISON ON HYPERNERF DATASET

We further provide a qualitative comparison among 4DGS (Wu et al., 2024), and USPLAT4D (ours) on the HyperNeRF (Park et al., 2021) dataset, as shown in Figure S2. By focusing on the dynamic, detail-rice regions, we observe that USPLAT4D restores noticeably finer details and sharper high-frequency structures. While 4DGS tends to oversmooth or lose small geometric components, our uncertainty-guided graph model consistently preserves these structures, leading to more stable and detailed 4D reconstructions.

## D ADDITIONAL ABLATION AND ANALYSIS

### D.1 ABLATION STUDY ON KEY / NON-KEY RATIO.

We select the uncertainty threshold by keeping the top 2% (1000-th) of all Gaussians. This strategy ensures broad spatial coverage while filtering unreliable nodes. We find this setting to be robust across scenes and base models. To further evaluate the impact of the threshold, we conduct an ablation study with different thresholds. As shown in Table S6, the results remain stable, and our method is not sensitive to this parameter. We vary the ratio from 0.005 to 0.04 and observed consistent performance, including our chosen value of 0.02, which demonstrates that using this well-justified parameter is reasonable and effective.

Table S6: **Ablation study on the uncertainty thresholds.** Results are reported relative to our chosen ratio threshold (0.02).

| % of Key Gaus. | Relative PSNR↑ | Relative SSIM↑ | Relative LPIPS↓ |
|---|---|---|---|
| MoSca (baseline) | 0.994 | 0.995 | 1.024 |
| 0.005 | 1.000 | 1.000 | 1.000 |
| 0.010 | 1.001 | 1.001 | 0.999 |
| 0.020 (reference) | 1.000 | 1.000 | 1.000 |
| 0.030 | 1.001 | 1.001 | 0.998 |
| 0.040 | 1.001 | 1.000 | 1.000 |

## D.2 ABLATION STUDY ON COLOR THRESHOLD

Table S7: **Ablation study on the color threshold $\eta_c$ using Objaverse dataset.** Results are reported relative to our chosen threshold ($\eta_c = 0.50$). $\eta_c = 1.00$ means no thresholding.

| $\eta_c$ | Relative PSNR↑ | Relative SSIM↑ | Relative LPIPS↓ |
|---|---|---|---|
| MoSca (baseline) | 0.980 | 0.992 | 1.115 |
| 0.01 | 0.993 | 0.999 | 1.023 |
| 0.10 | 0.996 | 1.000 | 1.011 |
| 0.20 | 0.994 | 1.000 | 1.014 |
| 0.40 | 0.998 | 1.000 | 1.008 |
| 0.50 (reference) | 1.000 | 1.000 | 1.000 |
| 0.80 | 1.001 | 1.000 | 1.001 |
| 1.00 | 0.997 | 1.000 | 1.010 |

The color threshold $\eta_c$ (see Equation 4) is introduced to prevent incorrect uncertainty estimation from Equation 3 when the prior Gaussians have not yet converged on certain pixels in the input images. We further study the influence of $\eta_c$ on the Objaverse dataset. As reported in Table S7, performance degrades when $\eta_c$ is set to $1.0$, which corresponds to disabling the threshold entirely. In this case, the model incorrectly trusts Gaussians with large color errors and assigns them falsely low uncertainty. On the other hand, when $\eta_c$ is reduced below $0.1$, many genuinely low-uncertainty Gaussians are mistakenly treated as unreliable, which again harms reconstruction quality. Although these two extremes lead to worse performance, we find a broad performance plateau between $[0.4, 0.8]$, showing that the hyperparameter is easy to select and not sensitive in practice. Across all tested values of $\eta_c$, our approach consistently outperforms MoSca.

## D.3 ABLATION STUDY ON THE SIGNIFICANT PERIOD THRESHOLD

As discussed in Section 4.2 of the main paper, we use a hyper-parameter (namely, Significant Period Threshold or SPT) to filter out Gaussians that are well observed for only a short duration (no more than four frames). Such transient Gaussians contribute little to motion propagation and often lead to weak or unreliable graph connections. In our experiments, we uniformly set the SPT to 5 for all datasets. To further assess its influence, we evaluate several values of SPT, as shown in Table S8. We observe a broad performance plateau when SPT $\geq 3$, indicating that the model is stable across a reasonable range of choices. Performance drops when the threshold is set to 1 (i.e., disabling the filtering by SPT) which introduces short-lived Gaussians into the key graph, which matches the intuition described above. For all choices of SPT, our method consistently achieves higher performance than MoSca across all tested values.

## D.4 ABLATION STUDY ON THE SCALING FACTOR

The scaling factor $[r_x, r_y, r_z]$ controls the relative weight when transforming 2D uncertainty into axis-aligned 3D uncertainty. The key insight is that the reliability of depth varies with camera motion. In scenarios where the camera undergoes large translation in the $x$-$y$ plane, the depth becomes better constrained. In such cases, down-weighting the depth component (using a smaller $r_z$) reduces noise in the uncertainty estimate. This intuition aligns with geometric models such as Direct Sparse Odometry

Table S8: Ablation study on the Significant Period (SPT). Results are reported relative to our chosen significant period threshold (SPT = 5).

| SPT | Relative PSNR↑ | Relative SSIM↑ | Relative LPIPS↓ |
|---|---|---|---|
| MoSca (baseline) | 0.980 | 0.992 | 1.109 |
| 1 (i.e., no thresholding) | 0.994 | 0.999 | 1.016 |
| 3 | 0.998 | 1.000 | 1.006 |
| 5 | 1.000 | 1.000 | 1.000 |
| 7 | 0.996 | 1.000 | 1.007 |
| 10 | 0.996 | 1.000 | 1.004 |

Table S9: **Ablation study on the depth uncertainty scale ratio** $r_z$. Results are reported relative to our chosen ratio threshold (1.00).

| $r_z$ or Baseline | Relative PSNR↑ | Relative SSIM↑ | Relative LPIPS↓ |
|---|---|---|---|
| MoSca (baseline) | 0.989 | 0.992 | 1.100 |
| 0.01 | 1.001 | 0.999 | 1.005 |
| 1.00 (ours, reference) | 1.001 | 1.000 | 1.000 |

(Engel et al., 2017), which models depth estimates with a variance term that is inversely proportional to the baseline, i.e., $\Delta Z \propto \frac{1}{b}$, where $\Delta Z$ is depth standard deviation and $b$ is camera baseline or translation.

Our design principle is that the default setting $[1, 1, 1]$ is valid, and once the camera motion is unbalanced across directions, adjusting the weights can improve robustness. For all experiments, we set $[r_x, r_y, r_z] = [1, 1, 0.01]$. We also evaluated other values of $r_z$ and did not observe notable differences (a small value works consistently). For datasets such as DyCheck, DAVIS, and NVIDIA, which have limited camera movement along the depth axis and often emphasize rotation or small shifts, using a smaller $r_z$ improves the stability of the uncertainty estimation. For objaverse, where the camera moves across all three spatial directions, adjusting $r_z$ from 1.0 to 0.01 has minimal side effect, as shown Table S9.

Table S10: **Ablation for comparable time on DyCheck dataset.**

| Setup | PSNR↑ | SSIM↑ | LPIPS↓ |
|---|---|---|---|
| MoSca (baseline) | 19.32 | 0.706 | 0.264 |
| Ours (same time with MoSca) | 19.41 | 0.710 | 0.254 |
| Ours (full time) | **19.63** | **0.715** | **0.249** |

### D.5 ANALYSIS ON RUNTIME

We provide a detailed runtime analysis (all measured on a single NVIDIA H100 GPU): (1) Uncertainty estimation: since the rendering speed is fast ($> 60$ FPS), this step is efficient and introduces negligible overhead. (2) Graph construction: ~3 sec/image. For a typical input sequence of 200 images in DyCheck (for DAVIS, it ranges 50~90 frames), this totals around 10 minutes. (3) Training time: ~4 sec/image, leading to 13 minutes for 200 input images. Importantly, our method does not require a fully optimized base model, which. To assess cost-effectiveness, we run USPLAT4D on an under-trained base model for the same total time as a fully trained baseline and observe comparable performance as shown in Table S10. Although the performance drops compared with our full model, it still have better performance than MoSca training with the same time.

To better understand how the time usage increases with video length, we further curate training scenes with different sequence lengths from the Objaverse dataset. Figure S3 shows the runtime of MoSca and our method relative to the input video length. The runtime of our method exhibits a smaller slope in both stages compared with MoSca.

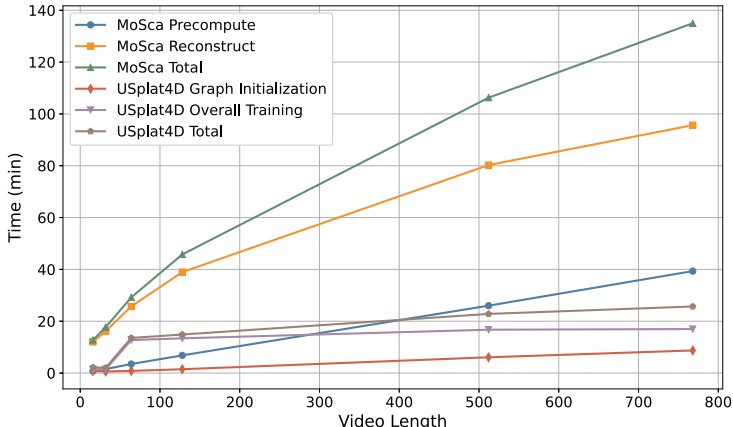

Figure S3: **Time usage.** MoSca involves a preprocessing stage followed by a reconstruction stage. *MoSca Total* is the sum of *MoSca Precompute* and *MoSca Reconstruct*. With the Gaussian priors from MoSca, our method includes a graph initialization stage (which covers uncertainty estimation and graph construction) and an overall training stage. *USplat4D Total* is the sum of *Graph Initialization* and *Overll Training*.

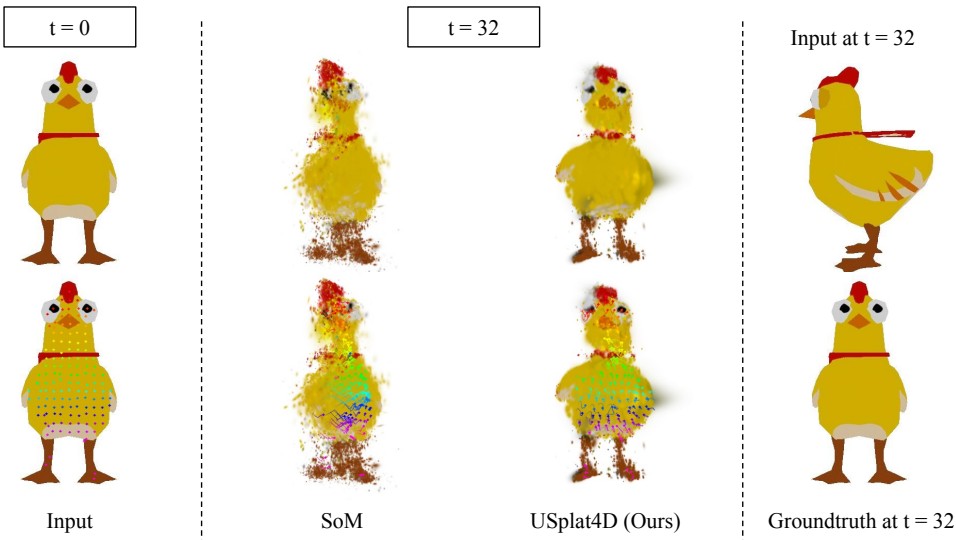

Figure S4: **Failure of tracking on textureless surface.** The input camera moves in a circular trajectory around a chick that remains quasi-static. We show the tracking and reconstruction results at frame 32 for both SoM and USPLAT4D (ours). Because the chick's surface lacks texture, the tracks sampled at the initial frame (t = 0) drift and accumulate incorrect motion under SoM, eventually causing the reconstruction to collapse. In contrast, our method is able to partially recover the geometry and produce more stable tracking.

# E    ANALYSIS OF CHALLENGING CASES

Although our model partially relies on the tracking quality of the initial 4D Gaussians and the geometric cues available during their observation period, both the quantitative and qualitative results demonstrate strong robustness in restoring or preserving shape under partial visibility and imperfect tracking. The uncertainty-guided graph allows the model to downweight unreliable motion and propagate stable cues across space and time. However, the framework still inherits the fundamental limitations of the underlying 4D prior. When the prior does not contain any reliable motion or geometric information, the uncertainty model has no meaningful signal to propagate, and therefore cannot fully recover the missing structures. Here, we further examine the impact of most common challenging cases: textureless regions, fast motion, and deformable objects.

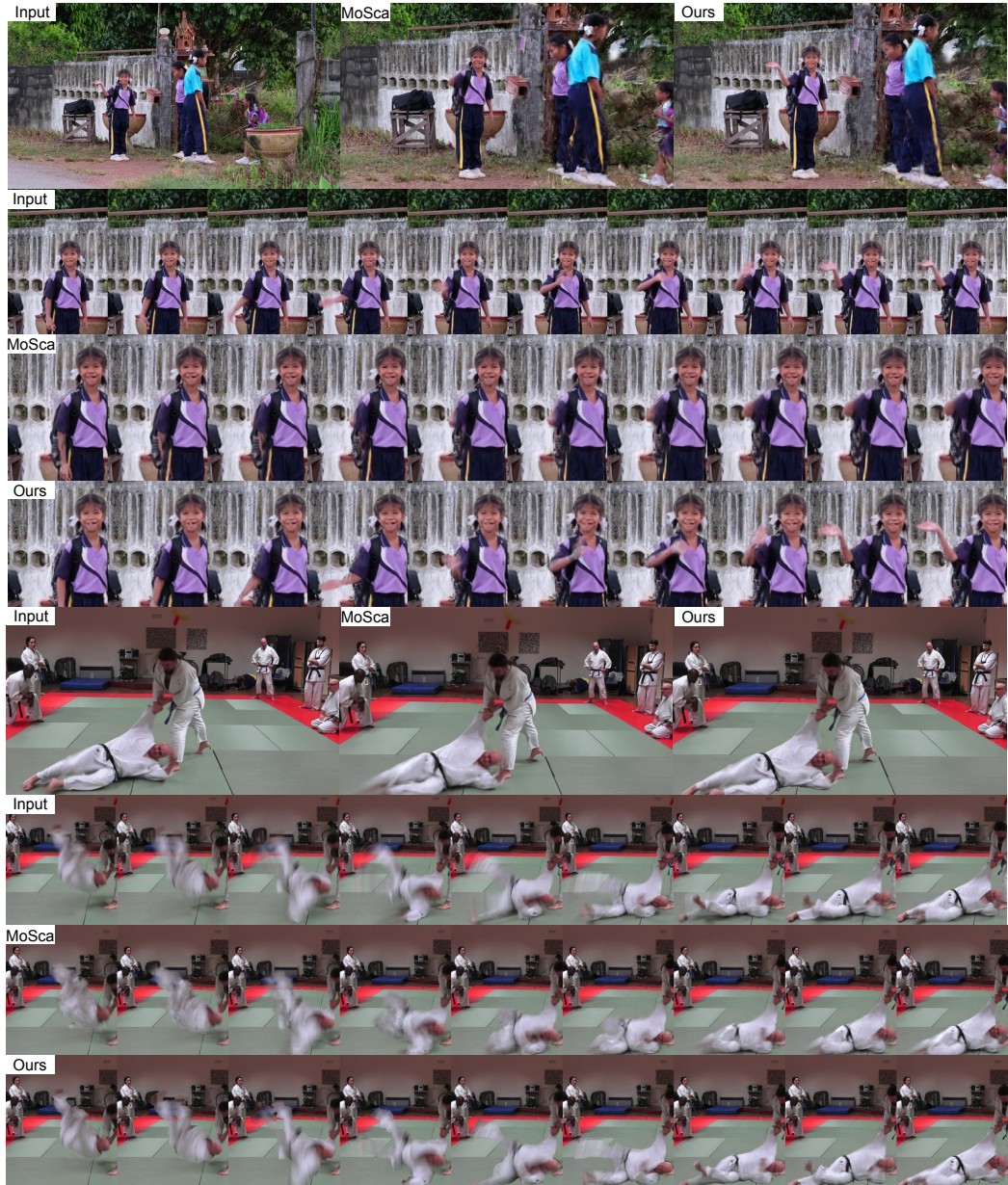

Figure S5: **Comparison of reconstruction results on fast-motion objects.** For each scene, the top row shows the overall view rendered from input views and novel views, and the three rows below highlight the fast-motion regions across ten successive time frames.

### E.1 TEXTURELESS REGION

In textureless areas, the vision foundation model produces unreliable tracks, which in turn causes SoM and MoSca to generate incorrect dynamic Gaussian priors. When our model receives these flawed priors, the uncertainty-guided graph attempts to propagate geometric information under the guidance of uncertainty across time and space. However, when the underlying correct information is rare, the propagated motion inevitably suffers from the poor initial tracking and reconstruction. In the Objaverse experiments, such textureless regions noticeably reduce quantitative performance compared to real-world datasets with richer appearance cues. Under our circling-camera setup, a typical failure case is shown in Figure S4. The Gaussians from SoM drift with the camera instead

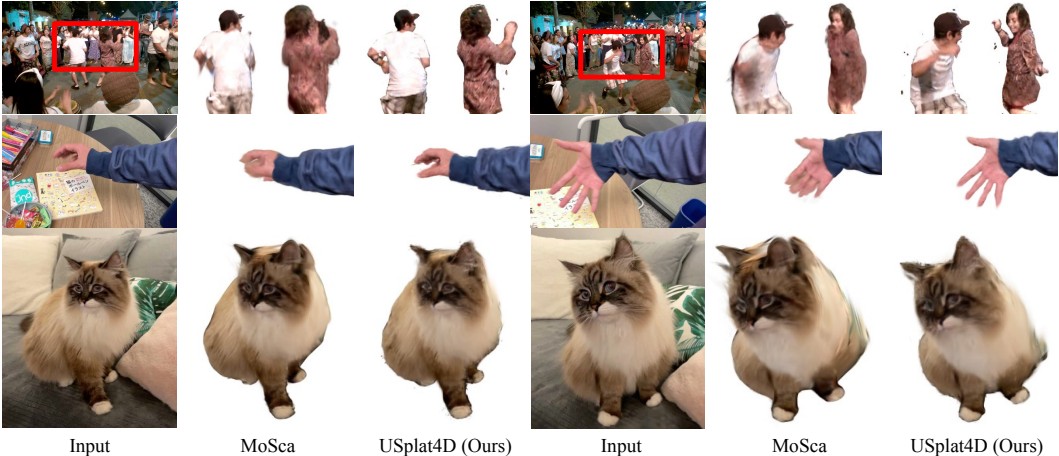

Figure S6: **Qualitative on deforming objects.** First row (Davis dancing scene): The subject's arms exhibit strong non-rigid motion throughout the dance. Second row (DyCheck handwavy scene): The hand undergoes continuous pose changes with significant articulation. Third row (DyCheck mochi-high-five scene): The cat's head rotates substantially. MoSca often produces drifting geometry where the ears or back of the neck separate from the head, while our method preserves consistent structure across time and viewpoints.

of representing stable geometric structures, while our method stabilizes the overall motion but still inherits errors in local regions where tracking is fundamentally unreliable.

### E.2 FAST MOTION

Monocular 4D reconstruction under fast motion remains highly challenging, yet our method demonstrates strong robustness in such scenarios. By employing an uncertainty-weighted DQB loss together with trainable motion parameters $\mathbf{p}_{i,t}$, the model can deviate from node-based interpolation when uncertainty signals unreliable priors, enabling recovery of local fast motion even under incorrect initialization (Figure S5). This capability arises from relaxing the hard constraints imposed by node-based interpolation. In contrast, existing monocular 4D Gaussian splatting approaches, such as SoM and MoSca, rely heavily on motion regularizers that favor smooth, temporally coherent trajectories, implicitly biasing them toward slow or moderate motion. Their motion models further limit fast dynamics: SoM restricts motion to a small set of bases, capping allowable local deformation, while MoSca interpolates trajectories via DQB on scaffolds, producing stable but nearly rigid motion. When scaffolds are missing in high-motion regions, this interpolation becomes a hard constraint that prevents Gaussians from following rapid, input-aligned motion. Despite these improvements, our reconstructions in fast-motion regions still inherit rolling-shutter distortions from the input video and rapid motion further challenges the preservation of multi-view consistency. Addressing such artifacts remains an open problem in monocular 4D reconstruction and is beyond the scope of this work.

### E.3 DEFORMING OBJECTS

Considering that deformable objects pose greater challenges than near-rigid ones, we conduct a qualitative evaluation of monocular 4D reconstruction in highly deforming scenarios. Figure S6 compares novel-view synthesis results of our method and MoSca across several subjects. Our reconstructions align more faithfully with the underlying geometry, particularly in regions with large non-rigid motion. This robustness stems from our uncertainty-guided spatio-temporal graph and uncertainty-weighted DQB loss, which softly regularize non-key nodes while allowing deviations when priors are unreliable. Consequently, even with imperfect Gaussian initialization from MoSca, our framework recovers fine-scale deforming geometry while maintaining spatio-temporal consistency. In contrast, MoSca interpolates all Gaussians using scaffold-based skinning weights. While effective with sufficient scaffold coverage, this design becomes brittle when scaffolds fail to capture corners, extremities, or thin structures due to hyperparameter sensitivity or inaccurate 2D priors. Gaussians in these regions become overly constrained, leading to distortions or missing parts, as seen in highly

deformable structures such as human limbs or the cat's ear in Figure S6. Despite these improvements, reconstructing highly deformable regions remains challenging when priors are severely inaccurate or observations are sparse, indicating directions for future work.

## F  SOCIAL AND BROADER IMPACT

Our work advances dynamic 3D scene reconstruction and novel view synthesis from monocular videos, with potential applications in AR/VR, physical scene understanding, digital content creation, and human-computer interaction. By modeling uncertainty and improving synthesis under extreme viewpoints, our method contributes to more robust and accessible 4D modeling. While the method has positive use cases, it could be misused for synthetic content manipulation. Our method does not involve sensitive data, and care should be taken in downstream applications to ensure responsible use.

