# OpenReview forum: "Uncertainty Matters in Dynamic Gaussian Splatting for Monocular 4D Reconstruction"
_ICLR.cc/2026/Conference — ICLR 2026 Poster_

### Official Review · Reviewer_SfVo · 2025-10-30

**Soundness:** 3
**Presentation:** 2
**Contribution:** 2
**Rating:** 6
**Confidence:** 4

**Summary:**

This paper presents a novel framework for monocular 4D scene reconstruction, emphasizing the critical role of uncertainty estimation in dynamic Gaussian splatting. The authors address key challenges in reconstructing dynamic scenes from monocular videos, particularly under occlusions and extreme viewpoint changes, by explicitly modeling the uncertainty associated with each Gaussian component in the scene. The proposed approach constructs an uncertainty-aware graph that guides the selection of reliable scene components (key nodes) and propagates motion information accordingly, resulting in improved geometric consistency and rendering quality.

**Strengths:**

The primary contribution of explicitly modeling and utilizing uncertainty within the scene reconstruction process represents a significant advancement. This approach effectively addresses a persistent challenge in monocular 4D reconstruction—robustly managing occlusions and view-dependent ambiguities.

The paper presents a systematic development of the framework, encompassing uncertainty estimation, graph construction, and optimization. It offers a comprehensive and cohesive pipeline that integrates seamlessly with existing Gaussian splatting methods. The ablation studies convincingly demonstrate the importance of each component, underscoring how uncertainty modeling enhances reconstruction quality.

The authors validate their approach across multiple datasets, including challenging synthetic benchmarks designed to rigorously test robustness. Both quantitative metrics and qualitative results substantiate that USPLAT4D delivers superior performance, especially in maintaining scene details from extreme viewpoints.

**Weaknesses:**

The effectiveness of the proposed framework hinges critically on the precision of the uncertainty estimation. Any inaccuracies in this component could propagate through the graph construction and optimization stages, potentially compromising the stability and fidelity of the reconstructed scene.

While the paper demonstrates significant improvements under various conditions, it provides limited discussion on scenarios where the uncertainty modeling may underperform. A more comprehensive analysis of failure modes—such as highly textureless regions or rapidly changing scenes—would strengthen the robustness claims.

Incorporating per-Gaussian uncertainty estimation and uncertainty-guided graph optimization introduces additional computational complexity. The paper lacks detailed analysis regarding the impact on processing time, resource requirements, and scalability to larger or real-time applications.

**Questions:**

How does the uncertainty-guided graph approach specifically improve robustness in occluded or fast-moving regions? Are there quantitative metrics or qualitative assessments that distinctly highlight these enhancements?

What is the added computational cost of incorporating the uncertainty estimation and graph optimization steps? Are there insights into throughput or latency impacts, especially for real-time or large-scale scenarios?

Did the experiments include scenarios with severe occlusion, sparse observations, or highly ambiguous regions? How does the model's uncertainty estimation perform under these challenging conditions?

---

> ### Author Response · Authors · 2025-11-24
> **Rebuttal 1/4**
>
> We thank the reviewer for their valuable comments.
>
> > **The effectiveness of the proposed framework hinges critically on the precision of the uncertainty estimation. Any inaccuracies in this component could propagate through the graph construction and optimization stages, potentially compromising the stability and fidelity of the reconstructed scene.**
>
> We derive a closed-form uncertainty under the assumption that training-view renderings are close to ground truth (local minima). Recognizing that this assumption may fail (e.g., at early training stages or in poorly constrained regions), we introduce an indicator-based fallback (Eq. (6) in the main paper) that assigns high uncertainty to unconverged Gaussians using color-error threshold. We then propagate the scalar uncertainty into a 3D anisotropic uncertainty matrix, capturing heterogeneous spatial reliability along image-plane and depth directions. This depth-aware formulation allows the model to reason about uncertainty in a geometrically meaningful way.
>
> Our careful design leads to consistent performance gains over SoM and MoSca across all evaluated datasets (DyCheck, DAVIS, Objaverse, NVIDIA). Even in the tracking failure case of the “chick” sequence (shown as Fig. S2 in the Appendix), where the uncertainty estimates are imperfect, the uncertainty-guided graph still significantly improves reconstruction compared to the baseline. We kindly invite the reviewer to examine Fig. S2 in the Appendix to see this behavior.
>
> That said, we fully agree that there is room for more advanced uncertainty modeling (e.g., learned uncertainty or multi-modal uncertainty). We view our current framework as a principled baseline and a solid stepping stone for future works on uncertainty in 4D Gaussian Splatting.
>
>
> > **While the paper demonstrates significant improvements under various conditions, it provides limited discussion on scenarios where the uncertainty modeling may underperform. A more comprehensive analysis of failure modes—such as highly textureless regions or rapidly changing scenes—would strengthen the robustness claims.**
>
> In Section D.7 and video demo of the original submission, we have discussed the challenging and failure cases of our method. Here, we provide further analysis.
>
> **Textureless regions.** As in Section D.7.1 and Fig. S2 of the appendix, textureless regions create unreliable 2D tracks, which in turn lead SoM and MoSca to produce incorrect dynamic Gaussian priors. When our method receives such flawed priors, the uncertainty-guided graph continues to propagate geometric information across time and space. While the propagated motion inevitably reflects the limitations of the initial tracking and reconstruction, our method stabilizes the global motion better than the original 4D priors.
>
> **Rapidly changing scenes.** Monocular 4D reconstruction under fast motion is a known challenge. Our method uses an uncertainty-weighted DQB loss together with trainable parameters $\mathbf{p}_{i,t}$ in Eq. 11 to represent the final motion. This design allows the model to deviate from scaffold-based interpolation in MoSca wherever uncertainty indicates unreliable priors, enabling recovery of local fast motion. Fig. S3 illustrates this behavior: MoSca fails to track rapid deformation when scaffolds are missing, while USplat4D still recovers approximate motion even when initialized with imperfect priors. Nevertheless, our reconstructions in fast-motion regions still inherit rolling-shutter distortions present in the input video. Fully resolving such camera-induced artifacts remains an open challenge in monocular 4D reconstruction.

---

> ### Author Response · Authors · 2025-11-24
> **Rebuttal 2/4**
>
> > **Incorporating per-Gaussian uncertainty estimation and uncertainty-guided graph optimization introduces additional computational complexity. The paper lacks detailed analysis regarding the impact on processing time, resource requirements, and scalability to larger or real-time applications.**
>
> > **What is the added computational cost of incorporating the uncertainty estimation and graph optimization steps? Are there insights into throughput or latency impacts, especially for real-time or large-scale scenarios?**
>
> **Detailed runtime analysis.**
> We have provided a runtime analysis in the original submission (see Section D.6 and Table S10 in the revised draft), which shows that USplat does not introduce too much overhead. Here, we further justify this and describe potential extensions for even larger scales.
>
> We provide a detailed runtime analysis (all measured on a single NVIDIA H100 GPU):
> (1) Uncertainty estimation: since the rendering speed is fast ($>$ 60 FPS), this step is efficient and introduces negligible overhead.
> (2) Graph construction: ~ 3 sec/image. For a typical input sequence of 200 images in DyCheck (for DAVIS, it ranges 50~90 frames), this totals around 10 minutes.
> (3) Training time: ~ 4 sec/image, leading to 13 minutes for 200 input images. Importantly, our method does not require a fully optimized base model, which. To assess cost-effectiveness, we run USplat4D on an under-trained base model for the same total time as a fully trained baseline and observe comparable performance as shown in Table S10. Although the performance drops compared with our full model, it still has better performance than MoSca training with the same time.
>
> **Scalability to long sequences.** We first report the measured time (in minutes) of MoSca and USplat4D components on a curated Objaverse dataset, where we can precisely control the video length. All measurements are on a single NVIDIA H100 GPU and include both precomputation and training components. Also, we kindly refer the reviewer to Fig. S1 in the appendix for better visualization.
> We observe that: (1) the MoSca total time increases with video length as expected; and (2) the additional cost introduced by USplat4D also grows with sequence length but remains significantly smaller than the MoSca training time (e.g., 25.68 min vs. 134.98 min at 768 frames). This supports our claim that USplat4D introduces a moderate and controlled overhead on top of strong 4D GS baselines.
>
> Table: Time consumption vs. video length on the curated Objaverse dataset.
> |Video length (#frames)|MoSca Precompute (min)|MoSca Reconstruct (min)|MoSca Total (min)|USplat4D Graph Initialization (min)|USplat4D Overall Training (min)|USplat4D Total (min)|
> |:--:|:--:|:--:|:--:|:--:|:--:|:--:|
> |16|0.73|12.00|12.73|0.77|1.42|2.18|
> |32|1.55|16.17|17.72|0.60|1.47|2.07|
> |64|3.48|25.72|29.20|0.83|12.72|13.55|
> |128|6.82|38.95|45.77|1.47|13.40|14.87|
> |512|26.00|80.27|106.27|6.08|16.73|22.82|
> |768|39.33|95.65|134.98|8.72|16.97|25.68|
>
> **Scalability to large scenes.**
>
> We agree with the reviewer that scaling to very large scenes is an important direction. Our current work focuses on the same target domain as SoM, MoSca, and similar monocular 4D GS models (DyCheck, DAVIS, NVIDIA, curated Objaverse), which contains scenes with multiple occluded objects.
> We analyzed multi-object scenes and showed results in Section D.7.4 and Fig. S5 in the Appendix. Our framework yields motion-based segmentation of Gaussians that closely aligns with per-object decomposition when multiple objects undergo distinct motions. We kindly invite the reviewer to look at these figures for visual examples of multi-object motion grouping and segmentation.
>
> Particularly, USplat treats each motion group as a separate “anchor cluster” via the graph structure. To justify this, in the Fig. S5 in the Appendix, we show that the key-graph weight matrix $\mathbf{W} ^{\mathrm{key}} = [w _{ij}] _{i,j \in \mathcal{V} _k}$ becomes approximately block-diagonal after reordering nodes using off-the-shelf graph community detection (e.g., spectral clustering). This reveals groups corresponding to distinct motion patterns (often aligning with object instances). Non-key nodes are then assigned to these motion groups via their connections in the non-key graph. In future work, we plan to make this explicit in the optimization formulation (e.g., group-wise regularization or group-specific hyperparameters) and explore settings with many interacting objects.
>
> **Real-time applications.** Our current implementation is not optimized for strict real-time applications; instead, we focus on offline high-quality reconstruction, similar to SoM and MoSca. We see improving throughput, latency, and scalability (e.g., via block-wise graphs and sliding-window processing) as an important direction for future work and will explicitly state this in the conclusion.

---

> ### Author Response · Authors · 2025-11-24
> **Rebuttal 3/4**
>
> > **How does the uncertainty-guided graph approach specifically improve robustness in occluded or fast-moving regions? Are there quantitative metrics or qualitative assessments that distinctly highlight these enhancements?**
>
> Our uncertainty-guided graph explicitly propagates reliable motion cues from well-observed Gaussians to those affected by occlusion, textureless surfaces, or fast motion.
>
> We estimate per-Gaussian uncertainty and then build a key / non-key graph. Key nodes (low uncertainty, frequently observed) serve as stable motion anchors; Non-key nodes (higher uncertainty or less frequent observations) are regularized by uncertainty-weighted motion propagation from nearby key nodes.
> When prior Gaussians from SoM or MoSca drift or collapse in occluded or fast-moving regions, our model aligns motion along directions where priors are reliable (low uncertainty) and allows deviations where priors are unreliable (high uncertainty).
>
> **Qualitative evidence.**  We explicitly evaluate on scenes with occlusion, deformation, textureless regions, and fast motion:
>
> Deforming subjects. In Fig. S4 in Appendix, we compare MoSca and our method on highly deformable objects (e.g., human arms/hands, animal ears). MoSca often produces distorted or missing structures in fast-moving or articulated parts, whereas our uncertainty-guided graph recovers more faithful geometry and yields temporally stable trajectories. We kindly invite the reviewer to examine Fig. S4 in the Appendix for these qualitative comparisons.
>
> Textureless regions. In Fig. S2, we show a typical failure case from Objaverse where textureless areas cause vision foundation models to produce unreliable tracks, leading SoM to drift with the camera and fail to represent stable structures. Our method stabilizes global motion and localizes errors to the intrinsically ambiguous regions. We kindly refer the reviewer to Fig. S2 for this example.
>
> Fast-motion regions. In Fig. S3, MoSca fails to track rapid deformation when scaffolds are missing in high-motion areas, leading to collapsed geometry. In contrast, our method couples an uncertainty-weighted DQB loss with trainable parameters $\mathbf{p}_{i,t}$ in Eq. 11, allowing the motion of non-key nodes to deviate from the behavior of scaffold-based interpolation (used in MoSca) for fast motion reconstruction. This recovers approximate motion even when the initialization is imperfect. We kindly invite the reviewer to look at Fig. S3 for visual evidence.
>
> **Quantitative robustness.**  Quantitatively, our experiments on DyCheck, Objaverse, and NVIDIA show that:
>
> Across all tested hyperparameter ranges (e.g., color threshold $\eta_c$ and significant period), our method consistently outperforms SoM and MoSca in PSNR/SSIM/LPIPS. Specifically, we curate the Objaverse dataset to conduct the evaluation on the extreme novel view to quantitatively analyze the occluded regions. Besides, we report the tracking performance in Table S2 in Appendix. The obvious improvement shows that our uncertainty-aware graph construction not only enhances visual quality but also improves spatio-temporal consistency for dynamic object tracking.
>
> Ablation studies (provided as tables in the Appendix and partially summarized in our responses to other reviewers) show broad performance plateaus for these hyperparameters, reflecting that the method remains stable under different levels of occlusion, sparse observations, and ambiguous regions.

---

> ### Author Response · Authors · 2025-11-24
> **Rebuttal 4/4**
>
> > **Did the experiments include scenarios with severe occlusion, sparse observations, or highly ambiguous regions? How does the model's uncertainty estimation perform under these challenging conditions?**
>
>
> Our experiments explicitly include cases with strong occlusion, fast motion, severely textureless regions, and ambiguous viewpoints:
>
> - DyCheck includes scenes with self-occlusion, large articulated deformation, and moderate-to-fast motion under real camera trajectories.
> - Davis contains challenging object-background interactions, complex motion, and occlusions.
> - Objaverse offers controlled synthetic scenes with textureless regions and extreme viewpoints (e.g., circling cameras, strong occlusion by the object itself).
> - NVIDIA provides dynamic scenes with limited camera motion but non-trivial object dynamics.
>
> Under these conditions, our uncertainty estimation behaves as follows:
>
> - Gaussians that are poorly constrained (e.g., only visible in very few views, heavily occluded, or lying in textureless regions) are assigned higher uncertainty, reducing their influence in the key graph.
> - Gaussians that are frequently and reliably observed receive lower uncertainty, and are therefore more likely to become key nodes and serve as anchors.
> - In fast-motion regions and regions with rolling-shutter distortions, we observe that uncertainty is elevated around Gaussians whose trajectories strongly disagree with local photometric evidence, which allows non-key nodes to deviate from scaffold-based interpolation.
>
> Empirically, this translates into:
>
> - More stable global motion in sequences where MoSca collapses or drifts under occlusion (visible in Fig. 1, Fig. 4, Fig.5 in the main paper).
> - Better preservation of thin or fast-moving structures (e.g., articulated limbs) where MoSca’s motion is over-smoothed or locked by missing scaffolds, as seen in Fig. S3 in the Appendix.
> - Consistent improvements in PSNR, SSIM, and LPIPS over SoM and MoSca on DyCheck, Davis, Objaverse, and NVIDIA, even when the prior tracking is heavily degraded.

---

### Official Review · Reviewer_57Ji · 2025-10-31

**Soundness:** 3
**Presentation:** 2
**Contribution:** 2
**Rating:** 4
**Confidence:** 3

**Summary:**

This work reconstruct 4D Gaussian Splatting for dynamic scenes from monocular videos. The proposed technique refines an initial 4DGS reconstructed by other method. They first classify each Gaussian into key and non-key nodes based on the uncertainty derived from rendering loss. Key nodes are finetuned to remain their original position and motion, while non-key nodes are regularized by the blended motion from nearby key nodes. Results on three datasets show improvement comparing to previous work.

**Strengths:**

I like the idea of to identify anchor Gaussians based on the rendering loss uncertainty. This may be useful for many other tasks which require some estimate of the importance of a Gaussian.

**Weaknesses:**

The original 4DGS seems to have much better quantitative and qualitative results from their paper, while the reported 4DGS looks much worse. For example, they report >30 PSNR on the synthetic dataset but the PSNR are all <20. I wonder the reason. Does the evaluated dataset in this paper exhibit some more challenge? Also, is will be even more convincing if the proposed method can be evaluated on the same set of datasets used in 4DGS.

I'm concern about the scalability of the graph pre-processing stage to scene with more Gaussians (e.g., larger scene, more complex scene, or long sequence). The existing graph pre-preprocessing already requires additional 10 more minutes. The computation and storage cost may increase even further and make it unpractical.

**Questions:**

Can the author provide some space and time complexity analysis for the graph pre-processing step (how they scale with the number of Gaussians)?

---

> ### Author Response · Authors · 2025-11-24
> **Rebuttal 1/3**
>
> We thank the reviewer for their valuable comments.
>
> > **The original 4DGS seems to have much better quantitative and qualitative results from their paper, while the reported 4DGS looks much worse. For example, they report >30 PSNR on the synthetic dataset but the PSNR are all <20. I wonder the reason. Does the evaluated dataset in this paper exhibit some more challenge? Also, is will be even more convincing if the proposed method can be evaluated on the same set of datasets used in 4DGS.**
>
> **Clarification on PSNR values.** We clarify all the baseline results including 4DGS in Table S1 are taken from the original SoM and MoSca papers. The discrepancy of PSNR values is from a dataset perspective. The D-NeRF benchmark used in the original 4DGS paper is synthetic, object-centric, and comes with ground-truth camera intrinsics and extrinsics. The scenes typically have: A single foreground object with slow or near-quasi-static motion; No cluttered background (often a uniform or simple background); Training views that densely cover the top hemisphere around the object, with test views largely within or very close to this trajectory. In contrast, the quantitative results we report in Table 1 and Table S1 are evaluated in the realistic DyCheck Dataset, which are substantially more challenging: Scenes contain complex, textured backgrounds and dynamic foreground objects with larger, more non-rigid motion. Camera calibration (intrinsics/extrinsics) is reconstructed and can be imperfect. Training images cover only a limited subset of views, and evaluation often includes novel views outside the input camera trajectory (e.g., extreme viewpoints in DyCheck and Objaverse).
> These differences in data difficulty naturally lead to lower PSNR values when evaluating only on truly dynamic regions and challenging viewpoints. This is consistent with other monocular dynamic NeRF/GS works evaluated on DyCheck-style settings.
>
> **Why we do not directly evaluate on the original 4DGS D-NeRF setup.** The D-NeRF-type synthetic datasets capturing slow-moving scene by an extremely-rapidly-moving (i.e., jumping) camera allows methods to leverage multi-view cues for optimization while the DyCheck dataset captures strictly-monocular scenarios, meaning that the camera is moving more naturally [A]. Our work specifically targets monocular video which align with most common real-world conditions (nature camera motion, complex backgrounds, imperfect calibration, extrapolative viewpoints) [B], with the same focus as the monocular 4DGS baselines such as SoM and MoSca. In contrast, the original 4DGS conduct experiments in non-strictly monocular synthetic dataset which capture images by the extremely-rapidly-moving cameras that rarely existed in the real world.
>
> We view 4DGS and USplat4D as complementary: 4DGS demonstrates what is achievable in well-calibrated, multi-view synthetic or controlled setups, whereas USplat4D focuses on uncertainty-aware refinement in monocular, real-world, and extrapolative scenarios. That said, adapting our method to the original 4DGS/D-NeRF evaluation is an interesting direction for future work, and we will clarify this scope difference explicitly in the main paper.
>
> [A] Liang, Yiqing, et al. "Monocular dynamic Gaussian splatting is fast and brittle but smooth motion helps." arXiv e-prints (2024): arXiv-2412.
>
> [B] Gao, Hang, et al. "Monocular dynamic view synthesis: A reality check." Advances in Neural Information Processing Systems 35 (2022): 33768-33780.

---

> ### Author Response · Authors · 2025-11-24
> **Rebuttal 2/3**
>
> > **I'm concern about the scalability of the graph pre-processing stage to scene with more Gaussians (e.g., larger scene, more complex scene, or long sequence). The existing graph pre-preprocessing already requires additional 10 more minutes. The computation and storage cost may increase even further and make it unpractical.**
>
> We have provided a runtime analysis in the original submission (see Section D.6 and Table S10 in the revised draft), which shows that USplat does not introduce too much overhead. Here, we further justify this and describe potential extensions for even larger scales.
>
> **Time analysis on long sequence.** We first report the measured time (in minutes) of MoSca and USplat4D components on a curated Objaverse dataset, where we can precisely control the video length. All measurements are on a single NVIDIA H100 GPU and include both precomputation and training components. Also, we kindly refer the reviewer to Fig. S1 in the appendix for better visualization.
> We observe that: (1) the MoSca total time increases with video length as expected; and (2) the additional cost introduced by USplat4D also grows with sequence length but remains significantly smaller than the MoSca training time (e.g., 25.68 min vs. 134.98 min at 768 frames). This supports our claim that USplat4D introduces a moderate and controlled overhead on top of strong 4D GS baselines.
>
>   Table: Time consumption vs. video length on the curated Objaverse dataset.
>   | Video length (#frames) | MoSca Precompute (min) | MoSca Reconstruct (min) | MoSca Total (min) | USplat4D Graph Initialization (min) | USplat4D Overall Training (min) | USplat4D Total (min) |
>   |:----------------------:|:----------------------:|:-----------------------:|:-----------------:|:-----------------------------------:|:-------------------------------:|:---------------------:|
>   | 16                     | 0.73                   | 12.00                   | 12.73             | 0.77                              | 1.42                            | 2.18                  |
>   | 32                     | 1.55                   | 16.17                   | 17.72             | 0.60                              | 1.47                            | 2.07                  |
>   | 64                     | 3.48                   | 25.72                   | 29.20             | 0.83                              | 12.72                           | 13.55                 |
>   | 128                    | 6.82                   | 38.95                   | 45.77             | 1.47                              | 13.40                           | 14.87                 |
>   | 512                    | 26.00                  | 80.27                   | 106.27            | 6.08                              | 16.73                           | 22.82                 |
>   | 768                    | 39.33                  | 95.65                   | 134.98            | 8.72                              | 16.97                           | 25.68                 |
>
> **Large, complex scenes.** We agree with the reviewer that scaling to very large scenes and very long sequences is an important direction. Our current work focuses on the same target domain as SoM, MoSca, and similar monocular 4D GS models (DyCheck, DAVIS, NVIDIA, curated Objaverse), which contains scenes with multiple occluded objects.
> We analyzed multi-object scenes and showed results in Section D.7.4 and Fig. S5 in the Appendix. Our framework yields motion-based segmentation of Gaussians that closely aligns with per-object decomposition when multiple objects undergo distinct motions. We kindly invite the reviewer to look at these figures for visual examples of multi-object motion grouping and segmentation.
>
> Particularly, USplat treats each motion group as a separate “anchor cluster” via the graph structure. To justify this, in the Fig. S5 in the Appendix, we show that the key-graph weight matrix $\mathbf{W}^{\mathrm{key}} = [w_{ij}]_{i,j \in \mathcal{V}_k}$ becomes approximately block-diagonal after reordering nodes using off-the-shelf graph community detection (e.g., spectral clustering). This reveals groups corresponding to distinct motion patterns (often aligning with object instances). Non-key nodes are then assigned to these motion groups via their connections in the non-key graph. In future work, we plan to make this explicit in the optimization formulation (e.g., group-wise regularization or group-specific hyperparameters) and explore settings with many interacting objects.

---

> ### Author Response · Authors · 2025-11-24
> **Rebuttal 3/3**
>
> > **Can the author provide some space and time complexity analysis for the graph pre-processing step (how they scale with the number of Gaussians)?**
>
> Let $N$ denote the total number of Gaussians, $T$ denote the number of frames, $\alpha \in [0, 1]$ be the fraction of key Gaussians (empirically $\approx 0.02$), so $|\mathcal{V}_k| = \alpha N$, and $K$ be the number of neighbors per node in the key graph (a fixed constant, e.g., $K=120$).
>
> Our preprocessing operates primarily on the key nodes rather than all Gaussians, which is crucial for scalability:
>
> For the key-node selection, at each frame, we evaluate uncertainty and visibility to select candidate key Gaussians. This involves per-Gaussian operations such as thresholding and counting the significant period. The cost scales linearly with the total number of Gaussian–frame pairs:
>   $$\mathcal{O}(N T).$$
>
> For the key-graph construction, We construct a kNN-style graph among key nodes. If we build a naive dense similarity matrix among key nodes at a particular reference time, the worst-case time complexity is
>   $$\mathcal{O}(|\mathcal{V}_k|^2) = \mathcal{O}(\alpha^2 N^2).$$
>   In practice, however, we use approximate neighbor search, and the fact that $\alpha \approx 0.02$ makes $|\mathcal{V}_k|$ much smaller than $N$. With kNN (and fixed $K$), the effective cost per frame is closer to
>   $$\mathcal{O}(|\mathcal{V}_k| \log |\mathcal{V}_k| + K |\mathcal{V}_k|),$$
>   so the overall graph construction across frames becomes
>   $$\mathcal{O}\big(T \cdot |\mathcal{V}_k| \log |\mathcal{V}_k|\big).$$
>
> For the Space complexity, the key graph stores at most $K$ outgoing edges per key node, so the adjacency structure requires
>   $$\mathcal{O}(K |\mathcal{V}_k|) = \mathcal{O}(K \alpha N)$$
>   space, which is linear in the number of Gaussians and independent of $T$ once the reference frame is chosen for graph construction.
>
> In summary, while the worst-case time complexity of building a dense similarity matrix over key nodes is quadratic in $|\mathcal{V}_k|$, the use of a small key-node ratio ($\alpha \approx 0.02$), kNN sparsification, and fixed $K$ keeps both the time and space requirements manageable in practice. Importantly, MoSca and many other dynamic 4D GS models (e.g., SoM) also incur $\mathcal{O}(n^2)$-like costs in their preprocessing (with respect to sequence length or Gaussians), so our asymptotic behavior aligns with existing baselines.

---

> ### Comment · Reviewer_57Ji · 2025-11-28
>
> Really appreciate Authors' gentle recap and the new analysis.
>
> My main concern about the evaluation is still remain. The main baseline 4DGS is not only evaluated on the pointed out synthetic D-NeRF dataset in their paper. They have also evaluated on the real-world HyperNeRF and Neu3D datasets, where their results look  much better quantitatively and qualitatively. Thus, I'm still not fully convinced by the reported 4DGS numbers on the datasets presented in this work. Reporting results on the same datasets as those in 4DGS is one way to fully address this concern.
>
> For now, I will not raise my rating due to the concern.

---

> > ### Author Response · Authors · 2025-12-03
> > **Additional Comparison with 4DGS**
> >
> > We sincerely thank the reviewer for carefully reading our first response.
> >
> > Per request, we further conduct the experiments using the HyperNeRF [A] dataset and show both the qualitative and quantitative results in Figure S6 in Appendix. We observe that our method restores noticeably finer details and sharper high-frequency structures. While 4DGS tends to oversmooth or lose small geometric components, our uncertainty-guided graph model consistently preserves these structures, leading to more stable and detailed 4D reconstructions.
> >
> > We would like to clarify that the 4DGS metrics on DyCheck are also cross-validated by Marbles [B] and MoSca, and we sincerely refer the reviewer to these accepted papers as independent confirmation that the reported 4DGS numbers are correct in our Table 1. To clarify why 4DGS behaves so differently across benchmarks, we briefly contrast D-NeRF (PSNR ≈ 34), HyperNeRF (PSNR ≈ 25), and DyCheck (PSNR < 15). On D-NeRF, 4DGS benefits from a non-strictly monocular setup utilizing multi-view cues and from full-frame evaluation dominated by large uniform backgrounds, which artificially boosts PSNR. On HyperNeRF, left and right camera images are alternated between training and validation, so evaluation views are extremely close in space and time to training views, making the task comparatively easy. In contrast, in the iPhone/DyCheck setting the input is strictly monocular, while evaluation uses static side cameras that are shifted away from the training trajectory. Although we described these as “relatively easy validation views” for USplat4D in the draft, this non-trivial shift forces 4DGS to perform much stronger extrapolation than in D-NeRF or HyperNeRF, which explains its significantly worse PSNR, SSIM, and LPIPS on DyCheck.
> >
> > Due to the new review process rules, we understand that we cannot further interact after this round. We nonetheless hope that these additional experiments on HyperNeRF and intuitive explanation help to solve this concern.
> >
> > [A] Park, Keunhong, et al. "Hypernerf: A higher-dimensional representation for topologically varying neural radiance fields." SIGGRAPH Asia 2021.
> >
> > [B] Stearns, Colton, et al. "Dynamic gaussian marbles for novel view synthesis of casual monocular videos." SIGGRAPH Asia 2024.

---

### Official Review · Reviewer_1czs · 2025-11-01

**Soundness:** 3
**Presentation:** 3
**Contribution:** 3
**Rating:** 4
**Confidence:** 4

**Summary:**

This paper presents USPLAT4D, a method for reconstructing 4D scenes from monocular video that explicitly models per-Gaussian, per-frame uncertainty and integrates it into the optimization process. The approach consists of three main components: (1) uncertainty estimation based on color reconstruction variance and convergence indicators, (2) uncertainty-aware k-NN (UA-kNN) graph construction with key/non-key node partitioning, and (3) unified optimization using strong anchor losses for key nodes and Dual Quaternion Blending (DQB) interpolation with uncertainty-weighted losses for non-key nodes.

**Strengths:**

1. Clear conceptual contribution: The paper articulates a compelling and principled insight—"frequently observed Gaussians serve as stable motion anchors"—and successfully integrates this into a quantitative optimization framework. The uncertainty-aware formulation is well-motivated.
2. Effective spatial-temporal propagation: The UA-kNN approach with key/non-key partitioning provides an elegant mechanism to propagate motion from reliable nodes to uncertain ones, effectively suppressing drift in occlusion scenarios.
3. Depth-aware uncertainty modeling: The propagation of 2D errors to 3D anisotropic covariances via camera rotation explicitly accounts for depth-directional uncertainty, addressing a practical issue (depth distortion) that many methods overlook.

**Weaknesses:**

1. Dependence on initialization: The method relies on pretrained dynamic Gaussian models (SoM/MoSca) for initialization. When initial tracking completely fails (e.g., textureless regions, extremely fast motion), uncertainty estimation alone cannot recover from poor initialization. While the authors acknowledge this, more analysis of failure modes would be valuable.
2. Hyperparameter dependencies: Multiple design choices affect performance (key/non-key ratio ~2%, significant period ≥5 frames, threshold η_c). The automatic adaptability to different scene characteristics appears limited, and more ablation studies on these choices would strengthen the paper.
3. Potential graph connectivity issues: The uncertainty-weighted edge formation may exclude low-uncertainty nodes, potentially missing important scene structures that are simply less frequently observed. This could lead to incomplete propagation in certain scenarios.
4. More fundamentally, this approach appears inherently limited to single-object, near-rigid scenarios. The core assumption—that frequently observed Gaussians serve as stable motion anchors—explains why results are strong on object-centric scenes like "paper-windmill," "spin," and Objaverse datasets, but raises concerns about applicability to scenes with independent motions or highly non-rigid deformations. The authors should explicitly acknowledge this as a scope constraint and provide analysis beyond curated object-centric benchmarks.

**Questions:**

- What modifications would be needed to handle scenes that don't fit the single-object, near-rigid assumption?
- Are there quantitative metrics that could predict when the method will fail based on scene characteristics (e.g., rigidity, object count, motion complexity)?

---

> ### Author Response · Authors · 2025-11-24
> **Rebuttal 1/4**
>
> We thank the reviewer for their valuable comments.
>
> > **Dependence on initialization: The method relies on pretrained dynamic Gaussian models (SoM/MoSca) for initialization. When initial tracking completely fails (e.g., textureless regions, extremely fast motion), uncertainty estimation alone cannot recover from poor initialization. While the authors acknowledge this, more analysis of failure modes would be valuable.**
>
> We acknowledge that the initialization matters. The goal of USplat4D is to correct imperfect priors via uncertainty-aware refinement. Here, we have added a detailed analysis of these failure modes in the Appendix, and we summarize them here.
>
> **Textureless regions.** As in Section D.7.1 and Fig. S2 of the appendix, textureless regions create unreliable 2D tracks, which in turn lead SoM and MoSca to produce incorrect dynamic Gaussian priors. When our method receives such flawed priors, the uncertainty-guided graph continues to propagate geometric information across time and space. While the propagated motion inevitably reflects the limitations of the initial tracking and reconstruction, our method stabilizes the global motion better than the original 4D priors.
>
> **Extremely fast motion.** Monocular 4D reconstruction under fast motion is a known challenge. Our method uses an uncertainty-weighted DQB loss together with trainable parameters $\mathbf{p}_{i,t}$ in Eq. 11 to represent the final motion. This design allows the model to deviate from scaffold-based interpolation in MoSca wherever uncertainty indicates unreliable priors, enabling recovery of local fast motion. Fig. S3 illustrates this behavior: MoSca fails to track rapid deformation when scaffolds are missing, while USplat4D still recovers approximate motion even when initialized with imperfect priors. Nevertheless, our reconstructions in fast-motion regions still inherit rolling-shutter distortions present in the input video. Fully resolving such camera-induced artifacts remains an open challenge in monocular 4D reconstruction.

---

> ### Author Response · Authors · 2025-11-24
> **Rebuttal 2/4**
>
> > **Hyperparameter dependencies: Multiple design choices affect performance (key/non-key ratio ~2%, significant period ≥5 frames, threshold η_c). The automatic adaptability to different scene characteristics appears limited, and more ablation studies on these choices would strengthen the paper.**
>
> **Key/non-key ratio (~2%).** We have discussed this in the origianl submission and we copy it here for your reference. We select key Gaussians by thresholding uncertainty such that we keep approximately the top 2% (around the 1000-th most confident Gaussian). This provides broad spatial coverage while filtering unreliable nodes. We evaluate the impact of this ratio by varying the percentage of key Gaussians from 0.5% to 4%. As shown below, performance remains stable, and our method is not sensitive to this parameter; the chosen value of 2% lies well within a broad plateau.
>
> |% of key Gaussians|Relative PSNR $\uparrow$|Relative SSIM $\uparrow$|Relative LPIPS $\downarrow$|
> |:--|:--:|:--:|:--:|
> |MoSca (baseline)|0.994|0.995|1.024|
> |0.005 (0.5%)|1.000|1.000|1.000|
> |0.01 (1.0%)|1.001|1.001|0.999|
> |0.02 (2.0%, ours, ref.)|1.000|1.000|1.000|
> |0.03 (3.0%)|1.001|1.001|0.998|
> |0.04 (4.0%)|1.001|1.000|1.000|
>
> Across all tested ratios, USplat4D consistently outperforms MoSca, and the variations between 0.5%–4% are minor. We have included this table as Table S5 in the Appendix.
>
> **Significant period (SP) threshold.** We use the SP threshold to filter out Gaussians that are well observed for only a short duration (e.g., one or two frames). Such transient Gaussians contribute little to motion propagation and often lead to weak or unreliable graph connections. In our experiments, we uniformly set $\textrm{SP} = 5$ frames for all datasets (roughly >0.4s for 10 fps video and >0.14s for 30 fps video).
>
> To assess its influence, we vary $\textrm{SP}$ and report results relative to $\textrm{SP} = 5$:
>
> Table: Ablation study on the SP threshold.
> | $\textrm{SP}$ or Baseline      | Relative PSNR $\uparrow$ | Relative SSIM $\uparrow$ | Relative LPIPS $\downarrow$ |
> |:--------------------------|:------------------------:|:------------------------:|:---------------------------:|
> | MoSca (baseline)          |          0.980           |          0.992           |            1.109            |
> | 1 (i.e., no thresholding) |          0.994           |          0.999           |            1.016            |
> | 3       |          0.998           |          1.000           |            1.006            |
> | 5             |          1.000           |          1.000           |            1.000            |
> | 7                         |          0.996           |          1.000           |            1.007            |
> | 10                        |          0.996           |          1.000           |            1.004            |
>
> We observe a broad performance plateau when $\textrm{SP} \geq 3$, indicating that the model is stable across a wide range. Performance drops when $\textrm{SP} = 1$, effectively disabling the filter and allowing very short-lived Gaussians into the key graph. For all $\textrm{SP}$ value, USplat4D consistently outperforms MoSca across all tested values. We have included this table as Table S8 in the Appendix.
>
> **Analysis of the threshold $\eta_c$.** The color threshold $\eta_c$ aims to prevent incorrect uncertainty estimation from Eq. (6) when prior Gaussians have not yet converged on certain pixels in the input images. It filters out pixels with clear color mismatch so that the uncertainty estimate remains stable.
>
> In practice, we find that USplat4D is robust once $\eta_c$ is set to a reasonable value. As shown below, a wide range of values in [0,1] leads to improvements over the MoSca baseline. When \eta becomes extremely large (larger than 1), the thresholding effect gradually weakens, which is expected, since nearly all pixels are then treated as converged. When \eta is too small, genuinely low-uncertainty Gaussians are filtered out. Across all experiments in the paper, we fix $\eta_c$ to 0.5. No dataset-specific tuning is needed. We updated these results in Section D.3 and Table S7 of the appendix.
>
>   Table: Ablation study on the color threshold $\eta_c$ on Objaverse.
> |$\eta_c$ or Baseline|Relative PSNR $\uparrow$|Relative SSIM $\uparrow$|Relative LPIPS $\downarrow$|
> |:--|:--:|:--:|:--:|
> |MoSca (baseline)|0.980|0.992|1.115|
> |0.01|0.993|0.999|1.023|
> |0.10|0.996|1.000|1.011|
> |0.20|0.994|1.000|1.014|
> |0.40|0.998|1.000|1.008|
> |0.50 (reference)|1.000|1.000|1.000|
> |0.80|1.001|1.000|1.001|
> |1.00 (i.e., no thresholding)|0.997|1.000|1.010|

---

> ### Author Response · Authors · 2025-11-24
> **Rebuttal 3/4**
>
> > **Potential graph connectivity issues: The uncertainty-weighted edge formation may exclude low-uncertainty nodes, potentially missing important scene structures that are simply less frequently observed. This could lead to incomplete propagation in certain scenarios.**
>
> We would like to clarify that edge construction in Eq. 7 and Eq. 8 follows the UA-kNN principle: edges are formed based on spatial distance modulated by time-aware uncertainty. In this formulation, a pair of Gaussians with low uncertainty and short spatial distance naturally yields a small UA-kNN metric, so these edges are always preserved. In other words, the edge formation does not exclude low-uncertainty neighbors that are spatially close. The only case in which an edge is omitted is when the Gaussians are far apart, which is the intended behavior of any kNN-based method.
>
> Second, the observation frequency of each Gaussian is captured by the significant period, which reflects how many frames a Gaussian remains reliably constrained. A short significant period indicates that the Gaussian is only briefly visible and therefore provides weaker long-term motion cues. However, the significant-period threshold influences only the key-node selection stage and does not affect edge formation among existing key nodes. Low-uncertainty Gaussians that are observed for only a short duration are simply not promoted to key nodes, but they remain fully included in the non-key graph. These Gaussians are still optimized throughout training and are guided by uncertainty-weighted DQB interpolation from neighboring key nodes. As a result, they continue to contribute to the reconstruction and are refined jointly with the rest of the model, rather than being discarded or ignored.
>
> Third, During the final training stage, we optimize all Gaussians together. Even if a Gaussian is not selected as a key node, its motion and geometry continue to be refined through its connections to reliable key nodes. This helps recover small geometry pieces and local motion that would otherwise collapse when the priors are unreliable or short-lived.
>
> > **More fundamentally, this approach appears inherently limited to single-object, near-rigid scenarios. The core assumption—that frequently observed Gaussians serve as stable motion anchors—explains why results are strong on object-centric scenes like "paper-windmill," "spin," and Objaverse datasets, but raises concerns about applicability to scenes with independent motions or highly non-rigid deformations. The authors should explicitly acknowledge this as a scope constraint and provide analysis beyond curated object-centric benchmarks.**
>
> We appreciate the reviewer’s insightful question that whether or not our model is limited to near-rigid scenarios, single-object. We want to clarify that our model can handle deformable objects and multiple objects. Here, we provide actual examples.
>
> **Deformable object.** We have added additional qualitative results in the updated Appendix for highly deformable scenes. Fig. S4 compares novel-view synthesis of MoSca and USplat4D on several deforming subjects (e.g., human arms/hands, animal ears). Overall, our reconstructions align more faithfully with the underlying geometry, particularly in regions undergoing large non-rigid motion.
>
> USplat4D incorporates an uncertainty-guided spatio-temporal graph and an uncertainty-weighted DQB loss that softly regularizes motion, allowing non-key nodes to deviate from interpolated motion when uncertainty indicates unreliable priors. However, MoSca represents motion using time-aware scaffolds and interpolates all Gaussians using associated skinning weights. This works well when scaffolds sufficiently cover the geometry, but it becomes brittle once scaffold placement fails to capture corners, extremities, or thin structures. In these cases, Gaussians in affected regions are overly constrained by the remaining scaffolds.
>
> **Multi-object scenes and motion segmentation.** We analyzed multi-object scenes and showed results in Section D.7.4 and Fig. S5 in the Appendix. Our framework yields motion-based segmentation of Gaussians that closely aligns with per-object decomposition when multiple objects undergo distinct motions. We kindly invite the reviewer to look at these figures for visual examples of multi-object motion grouping and segmentation.
>
>
> **Acknowledging remaining limitations.** We agree that extremely complex scenes with many independently moving objects, topological changes, or highly articulated deformations remain challenging. We also acknowledge that our current quantitative evaluation, which follows the standard SOTA dynamic NeRF/GS evaluation protocol (Table S1), is primarily conducted on object-centric and moderately complex scenes. Building benchmarks with more complex dynamic scenes and multiple validation views is an important and necessary direction in this research area for further assessing 4D reconstruction methods, including ours.

---

> ### Author Response · Authors · 2025-11-24
> **Rebuttal 4/4**
>
> > **What modifications would be needed to handle scenes that don't fit the single-object, near-rigid assumption?**
>
> As discussed above, our model can handle multiple and deformable objects. In multi-object settings, USplat4D treats each motion group as a separate “anchor cluster” via the graph structure. To justify this, in the Fig. S5 in the Appendix, we show that the key-graph weight matrix $\mathbf{W}^{\mathrm{key}} = [w_{ij}]_{i,j \in \mathcal{V}_k}$ becomes approximately block-diagonal after reordering nodes using off-the-shelf graph community detection (e.g., spectral clustering). This reveals groups corresponding to distinct motion patterns (often aligning with object instances). Non-key nodes are then assigned to these motion groups via their connections in the non-key graph. In future work, we plan to make this explicit in the optimization formulation (e.g., group-wise regularization or group-specific hyperparameters) and explore settings with many interacting objects.
>
> > **Are there quantitative metrics that could predict when the method will fail based on scene characteristics (e.g., rigidity, object count, motion complexity)?**
>
> We appreciate this question. We also acknowledge this is an open question. When designing USplat4D, we indeed used intermediate quantities as diagnostics to understand when the model may fail or when hyperparameters are poorly chosen. We will add a dedicated subsection in the Appendix to summarize these metrics. Below we highlight the main ones.
>
> **Sparsity of the key-graph weight matrix.**  For multi-object cases, we expect the key-graph weight matrix $\mathbf{W}^{\mathrm{key}}$ to be *approximately block-diagonal* after node reordering, reflecting distinct motion groups. When hyperparameters are extremely poor or the model is under-trained, this matrix becomes dense, indicating that different motion groups are not well separated. We quantify sparsity as:
> $$\text{loss} _{\text{sparsity}} = \frac{1}{N} \sum _{i=1} ^{N} \frac{\lVert w _i \rVert _{1}}{\lVert w _i \rVert _{2}}$$
> where $w_i$ is the $i$-th row of $\mathbf{W}^{\mathrm{key}}$. A large value suggests overly dense connections and potentially poor segmentation of motion groups. We kindly refer the reviewer to Fig. S5 in the Appendix, where the block structure of $\mathbf{W}^{\mathrm{key}}$ is visualized.
>
> **$\mathcal{L}^{\text{motion,key}}$ term (Eq. (9) in the main paper).**  We monitor the motion consistency loss on key nodes, $\mathcal{L}^{\text{motion,key}}$. When the 4D Gaussian prior from SoM/MoSca collapses under extreme novel views, we expect this term to decrease to very small values (e.g., below 1% of the initial loss), indicating that the key-node motion has stabilized. If $\mathcal{L}^{\text{motion,key}}$ remains high, it typically signals that extreme views still exhibit noticeable discrepancies and that either (a) the base priors are too unreliable or (b) the optimization has not converged.
>
> **Position drift of low-uncertainty Gaussians.**  USplat4D assumes that Gaussians with low uncertainty should serve as relatively stable anchors. When tracking and reconstruction fail and the failed regions are erroneously assigned low uncertainty, we observe that such Gaussians undergo large position changes even during their low-uncertainty period after optimization. This mismatch indicates a clear discrepancy between the uncertainty estimates and the underlying geometry.
>
> Formally, we can measure:
>
>  $$\sum _{t=0} ^{T-1} \sum _{i \in \mathcal{V}_k \cup \mathcal{V}_n} \lVert \mathbf{p} _{i,t} ^{\text{(init)}} - \mathbf{p} _{i,t} ^{\text{(final)}}\big \rVert _2,$$
>
> which focuses on Gaussians that were classified as low-uncertainty for a significant duration. We successfully use this positional drift metric to detect tracking failure cases such as Fig. S3 in the Appendix. Large drifts in supposed “anchor” Gaussians indicate potential failure.
>
> These metrics do not yet constitute a fully automatic failure predictor, but they provide quantitative indicators correlated with scene difficulty (motion complexity, quality of priors) and model reliability.

---

### Official Review · Reviewer_3BJu · 2025-11-02

**Soundness:** 3
**Presentation:** 3
**Contribution:** 3
**Rating:** 6
**Confidence:** 4

**Summary:**

This work introduces USplat4D, an uncertainty-aware dynamic Gaussian Splatting framework for monocular 4D scene reconstruction. The key idea is to explicitly estimate per-Gaussian, time-varying uncertainty and leverage it to organize Gaussians into a spatio-temporal graph. This uncertainty-aware graph informs the selection of reliable (“key”) and less reliable (“non-key”) nodes, guiding motion propagation and adaptive optimization for more robust geometry and novel view synthesis, especially under occlusions and extreme viewpoints. Experiments on several real and synthetic datasets demonstrate superior consistency, quality, and robustness versus state-of-the-art baselines in both quantitative and qualitative settings.

**Strengths:**

Strength:
1. The paper is well-written and easy to follow.
2. Usplat achieve good result against recent baselines.

**Weaknesses:**

Major Weakness:
1. The overall pipeline involves multiple stages and components, baseline reconstruction model Mosca is already complex, USplat makes it appear rather complex and add time complexity.
2. Usplat ommit a evaluation on the nvidia dynamic dataset.

Minor Weakness:
1. In line 60, The should be "the"

**Questions:**

1. How does the choice of the threshold $ \eta_c $ in Eq. 7, or the scaling constants $ (r_x, r_y, r_z) $ in Eq. 9, affect downstream geometry and rendering quality? How is the method robust to different settings?
2. Can you provide more analysis on the time consumption.

---

> ### Author Response · Authors · 2025-11-24
> **Rebuttal 1/3**
>
> We thank the reviewer for their valuable comments.
>
> > **The overall pipeline involves multiple stages and components, baseline reconstruction model Mosca is already complex, USplat4D makes it appear rather complex and adds time complexity.**
>
> We would like to clarify that our detailed pipeline description is intended to ensure full reproducibility and should not be interpreted as an attempt to complicate our method. The core idea of USplat4D is simple: we estimate uncertainty and use it to guide dynamic motion modeling. The graph construction and its optimization follow standard practice in many computer vision tasks and are not the source of any additional complexity. The technical value lies in how this graph is modeled, and USplat4D offers an effective way that improves dynamic reconstruction.
>
> We certainly agree with the reviewer that the existing 4D models are complex. The fact that (1) our solution yields substantial improvements and (2) many different 4D models can be plugged into our pipeline further justifies the feasibility and potential of USplat4D. As 4D models continue to progress, we expect 4D priors to be obtained in a simpler way, and USplat4D is positioned to function as a practical refinement module that can be attached to future systems.
>
> Regarding time, we have provided a runtime analysis in the original submission (see Section D.6 and Table S10 in the revised draft), which shows that USplat does not introduce too much overhead. In our response to the reviewer’s later question, we include a further analysis that supports this observation.
>
> > **Usplat omit an evaluation on the NVIDIA dynamic dataset.**
>
> We have included the NVIDIA results in Appendix C.4 and Table S4 in the original submission. For convenience, we reproduce Table S4 below. USplat4D improves performance when applied to state-of-the-art MoSca, although the gains are smaller than those observed on other benchmarks. This is expected, as the NVIDIA dataset contains input views with limited motion, while benchmarks with larger camera movement and more challenging dynamics (e.g., DyCheck, DAVIS, Objaverse) benefit more from our method, leading to more substantial improvements.
>
> Table S4: Evaluation on NVIDIA  dataset.
> | Method      | PSNR $\uparrow$ | SSIM $\uparrow$ | LPIPS $\downarrow$ |
> |-----------------|------------|------------|--------------|
> | MoSca           | 26.77      | 0.854      | **0.07**     |
> | **USplat4D**    | **26.93**  | **0.855**  | **0.07**     |
>
> > **In line 60, The should be "the".**
>
> Thank you, and we have fixed this typo.

---

> ### Author Response · Authors · 2025-11-24
> **Rebuttal 2/3**
>
> > **How does the choice of the threshold  in Eq. 7, or the scaling constants  in Eq. 9, affect downstream geometry and rendering quality? How is the method robust to different settings?**
>
> **Analysis of the color threshold $\eta_c$.** The color threshold $\eta_c$ aims to prevent incorrect uncertainty estimation from Eq. (6) when prior Gaussians have not yet converged on certain pixels in the input images. It filters out pixels with clear color mismatch so that the uncertainty estimate remains stable.
>
> In practice, we find that USplat4D is robust once $\eta_c$ is set to a reasonable value. As shown below, a wide range of values in [0,1] leads to improvements over the MoSca baseline. When $\eta_c$ becomes extremely large (larger than 1), the thresholding effect gradually weakens, which is expected, since nearly all pixels are then treated as converged. When $\eta_c$ is too small, genuinely low-uncertainty Gaussians are filtered out. Across all experiments in the paper, we fix $\eta_c$ to 0.5. No dataset-specific tuning is needed. We updated these results in Section D.3 and Table S7 of the appendix.
>
>   Table: Ablation study on the color threshold $\eta_c$ on Objaverse.
>
> | $\eta_c$ or Baseline            | Relative PSNR $\uparrow$ | Relative SSIM $\uparrow$ | Relative LPIPS $\downarrow$ |
> |:--------------------------------|:------------------------:|:------------------------:|:---------------------------:|
> | MoSca (baseline)                |          0.980           |          0.992           |            1.115            |
> | 0.01                            |          0.993           |          0.999           |            1.023            |
> | 0.10                            |          0.996           |          1.000           |            1.011            |
> | 0.20                            |          0.994           |          1.000           |            1.014            |
> | 0.40                            |          0.998           |          1.000           |            1.008            |
> | 0.50 (reference)                |          1.000           |          1.000           |            1.000            |
> | 0.80                            |          1.001           |          1.000           |            1.001            |
> | 1.00 (i.e., no thresholding)    |          0.997           |          1.000           |            1.010            |
>
> **Analysis on the scaling factor.** The scaling factor $[r_x, r_y, r_z]$ controls the relative weight when transforming 2D uncertainty into axis-aligned 3D uncertainty. The key insight is that the reliability of depth varies with camera motion. In scenarios where the camera undergoes large translation in the x-y plane, the depth becomes better constrained. In such cases, down-weighting the depth component (using a smaller $r_z$) reduces noise in the uncertainty estimate. This intuition aligns with geometric models such as Direct Sparse Odometry [A], which models depth estimates with a variance term that is inversely proportional to the baseline, i.e., $\Delta Z \propto \frac{1}{b}$, where $\Delta Z$ is depth standard deviation and $b$ is camera baseline or translation.
>
> Our design principle is that the default setting [1, 1, 1] is valid, and once the camera motion is unbalanced across directions, adjusting the weights can improve robustness. For all experiments, we set $[r_x, r_y, r_z]$ = [1, 1, 0.01]. We also evaluated other values of $r_z$ and did not observe notable differences (a small value works consistently). For datasets such as DyCheck, DAVIS, and NVIDIA, which have limited camera movement along the depth axis and often emphasize rotation or small shifts, using a smaller $r_z$ improves the stability of the uncertainty estimation. For objaverse, where the camera moves across all three spatial directions, adjusting $r_z$ from 1.0 to 0.01 has minimal side effect, as shown in our ablation study below.
>
>   Table: Ablation study on the depth scaling factor $r_z$ on Objaverse. Results are reported relative to $r_z=1$ (ours, reference).
>
>   | Method              | $r_z$   | Relative PSNR $\uparrow$ | Relative SSIM $\uparrow$ | Relative LPIPS $\downarrow$ |
>   |:--------------------|:-------:|:------------------------:|:------------------------:|:---------------------------:|
>   | MoSca (baseline)    |   –     |          0.989           |          0.992           |            1.100            |
>   | USplat4D (ours)     | 0.01   |          1.001           |          0.999           |            1.005            |
>   | USplat4D (ours)     | 1.0     |          1.000           |          1.000           |            1.000            |
>
>
> We have updated this discussion in Section D.5 and Table S9 of the appendix.
>
> [A] Engel, Jakob, Vladlen Koltun, and Daniel Cremers. "Direct sparse odometry." IEEE transactions on pattern analysis and machine intelligence 40.3 (2017): 611-625.

---

> ### Author Response · Authors · 2025-11-24
> **Rebuttal 3/3**
>
> > **Can you provide more analysis on the time consumption.**
>
> **Discussion in the original submission.** We have provided a runtime analysis in the original submission (see Section D.6 and Table S10 in the revised draft), which shows that USplat4D does not introduce too much overhead. Here, we restate the key result in the table below, using DyCheck as an example. Here “Ours (same time)” means that we limit the total training time of “MoSca + USplat4D” to be the same as the original MoSca training budget. Even when constrained to the same total time as MoSca, our method already provides better performance. With a moderate ~23.5% extra time, we obtain a more noticeable improvement in both PSNR/SSIM and LPIPS. We humbly think this additional cost is reasonable given the gains, especially under challenging viewpoints.
>
>   Table: Ablation for comparable time on DyCheck dataset.
>   | Setup                         | PSNR$\uparrow$ | SSIM$\uparrow$ | LPIPS$\downarrow$ |
>   |:------------------------------|:--------------:|:--------------:|:-----------------:|
>   | MoSca (baseline)             |     19.32      |     0.706      |       0.264       |
>   | USplat4D (same time as MoSca) |     19.41      |     0.710      |       0.254       |
>   | USplat4D (full time: +23.5%)  |     19.63      |     0.715      |       0.249       |
>
> **Runtime vs. video length.** To further analyze time consumption, we conduct experiments on a curated Objaverse dataset where we can precisely control the video length. Here, we focus solely on the runtime of MoSca and the additional time introduced by USplat4D (graph initialization + overall training), without involving performance metrics. All timings are measured on a single NVIDIA H100 GPU. We kindly refer the reviewer to Fig. S1 in the appendix for better visualization. We also attach the table version below for reference.
>
> We observe that: (1) the MoSca total time increases with video length as expected; and (2) the additional cost introduced by USplat4D also grows with sequence length but remains significantly smaller than the MoSca training time (e.g., 25.68 min vs. 134.98 min at 768 frames). This supports our claim that USplat4D introduces a **moderate and controlled overhead** on top of strong 4D GS baselines.
>
>   Table: Time consumption vs. video length on the curated Objaverse dataset.
>   | Video length (#frames) | MoSca Precompute (min) | MoSca Reconstruct (min) | MoSca Total (min) | USplat4D Graph Initialization (min) | USplat4D Overall Training (min) | USplat4D Total (min) |
>   |:----------------------:|:----------------------:|:-----------------------:|:-----------------:|:-----------------------------------:|:-------------------------------:|:---------------------:|
>   | 16                     | 0.73                   | 12.00                   | 12.73             | 0.77                              | 1.42                            | 2.18                  |
>   | 32                     | 1.55                   | 16.17                   | 17.72             | 0.60                              | 1.47                            | 2.07                  |
>   | 64                     | 3.48                   | 25.72                   | 29.20             | 0.83                              | 12.72                           | 13.55                 |
>   | 128                    | 6.82                   | 38.95                   | 45.77             | 1.47                              | 13.40                           | 14.87                 |
>   | 512                    | 26.00                  | 80.27                   | 106.27            | 6.08                              | 16.73                           | 22.82                 |
>   | 768                    | 39.33                  | 95.65                   | 134.98            | 8.72                              | 16.97                           | 25.68                 |

---

### Author Response · Authors · 2025-12-03
**Summary of Rebuttal for AC (1/2)**

Dear AC,

In light of the recent changes to this year’s ICLR review policy, we provide below a summary that (i) lists our paper’s contribution, (ii) reviewer assessments, and (iii) summarizes how each reviewer’s concerns were addressed in the rebuttal and revision.

---
## **Contributions**
**New problem focus**: we study a largely overlooked and under-explored aspect of 4D reconstruction: extreme novel view synthesis from monocular videos, such as opposite or side views that are far from the input trajectory. Current 4D models are evaluated mainly on viewpoints near the input, leaving this challenge insufficiently examined.

**New perspective and insight**: we identify that the key factor enabling such challenging view synthesis is to rely on the most reliable observations (frequently observed Gaussian points) and use them to guide motion estimation in dynamic scenes.

**Methodological novelty**: we propose USplat4D, which includes (1) an uncertainty estimation method that identifies reliable Gaussian points, and (2) an uncertainty-aware graph model that propagates their motion to support consistent dynamic modeling.

**Empirical significance**: extensive experiments across several multiple (DyCheck, DAVIS, Objaverse, NVIDIA, HyperNeRF) demonstrate that USplat4D yields substantial improvements over strong baselines. We provide detailed analysis and ablation studies to support the key design choices, as well as the effectiveness and efficiency of USplat4D.

---
## **Positive assessments**
All four reviewers recognized the motivation, novelty, clear contribution, technical soundness, and strong empirical results of our work. We are grateful they “like our idea” (57Ji); found “our conceptual contribution clear, compelling, and principled” (1czs), described our uncertainty-aware method “addressing a practical issues that many methods overlook” (1czs), “comprehensive” (SfVo), and “usefully for many other tasks” (57Ji); noted our improvement “significant” (SfVo) and “good” (3BJu); our evaluation and ablation “convincing” (SfVo).

At the time the discussion was frozen, our paper had the following scores:
- 3BJu: 6 (had not yet responded)
- 1czs: 4 (had not yet responded)
- 57Ji: 4 (responded but had not finished the discussion)
- SfVo: 6 (had not yet responded)

---

> ### Author Response · Authors · 2025-12-03
> **Summary of Rebuttal for AC (2/2)**
>
> ## **Addressing reviewer concerns**
>
> ---
> **[Reviewer 3BJu, score: 6]**
>
> Concerns raised:
> - Argued that the pipeline is complex.
> - Asked for results on the NVIDIA dataset.
> - Requested ablations on $\eta_c$ (Eq. 7) and scaling factors (Eq. 9).
> - Asked for time consumption analysis.
>
> How we addressed them:
> - Clarified that the detailed pipeline description is to ensure full reproducibility and should not be interpreted as attempting to complicate the method. The core idea is simple: estimate uncertainty and use it to guide dynamic motion.
> - Clarified that NVIDIA results were already included in the original submission (Appendix C.4).
> - Added ablations showing robustness to $\eta_c$ and the scaling factors (Appendix D.3, D.5).
> - Clarified that runtime analysis was already provided and added further evaluation on long and complex videos (Appendix D.6).
>
> ---
> **[Reviewer 1czs, score: 4]**
>
> Concerns raised:
> - Questioned challenging cases when initialization may fail, e.g., textureless regions, fast motion.
> - Requested ablations on the key/non-key ratio, significant period, and $\eta_c$ (Eq. 7).
> - Questioned whether graph construction may exclude low-uncertainty nodes.
> - Requested clarification on multi-object and non-rigid scenes.
>
>
> How we addressed them:
> - Acknowledged challenges and showed examples demonstrating that uncertainty-aware refinement corrects imperfect priors in textureless and fast-motion regions (Appendix D.7.1, D.7.2).
> - Clarified we had discussed key/non-key ration and significant period in the original submission; added ablation of $\eta_c$ (Appendix D.1, D.3, D.4).
> - Resolved a misunderstanding by explaining that the edge formulation does not exclude low-uncertainty neighbors and preserves important structures.
> - Provided examples of multi-object and non-rigid scenes, showing that the method handles them without modification (Appendix D.7.3, D.7.4).
>
> ---
> **[Reviewer 57Ji, score: 4]**
>
> Concerns raised:
> - Requested comparison with 4DGS.
> - Questioned scalability for large, complex, and long scenes.
> - Asked for space and time complexity analysis.
>
> How we addressed them:
> - Clarified that 4DGS comparisons was already included and followed prior work; added further results on the HyperNeRF dataset, where our method still outperformed (Appendix D.8).
> - Presented examples of large-scale and long-duration sequences (Appendix D.6).
> - Provided space and time complexity analysis consistent with baseline methods.
>
> ---
> **[Reviewer SfVo, score: 6]**
>
> Concerns raised:
> - Questioned the precision of uncertainty estimation.
> - Questioned performance in textureless regions and rapidly changing scenes.
> - Asked about computational cost.
>
> How we addressed them:
> - Clarified the theoretical basis and empirical reliability of the uncertainty estimation.
> - Presented actual cases in textureless and rapidly varying scenes where our method outperforms baselines (Appendix D.7.1, D.7.2).
> - Noted that runtime analysis was already included and added detailed results on long and large scenes (Appendix D.6), emphasizing that real-time performance is not the target.
>
> ---
> ## **Summary and final remarks**
> There is a clear consensus among all four reviewers regarding the idea, motivation, novelty, and significance of our uncertainty-aware perspective. During the rebuttal phase, we addressed every concern through additional experiments and clarification. Several questions (such as time analysis, NVIDIA results, and challenging cases) were already answered in the original submission.
>
> We emphasize that this work offers a fresh perspective for monocular 4D reconstruction. We identify how to detect reliable Gaussian points and use them to guide motion propagation, and we propose a principled uncertainty estimation method to support this idea. Regarding time complexity, we provided extensive analysis on long videos and complex scenes, together with detailed space and time complexity discussions. Empirical results across multiple datasets, along with **a 7-minute demo video** in supplementary material, clearly show substantial improvement over strong baselines.
>
> We believe our work highlights an important aspect of dynamic scene reconstruction: extreme novel view synthesis. We hope this study encourages more comprehensive evaluation of dynamic reconstruction models rather than focusing only on viewpoints near the input trajectory.
>
> We hope this summary provides a useful reference for the AC. Thank you for your time and consideration.

---

### Meta-Review · Area_Chair_twnC · 2025-12-23

**Summary:**

This paper proposes USplat4D, an uncertainty-aware dynamic Gaussian Splatting framework that propagates reliable motion cues to enhance 4D reconstruction. Uncertainty of Gaussians is estimated and a spatio-temporal graph is constructed for uncertainty-aware optimization.
The concerns of the reviewers include lack of analysis on computational cost, lack of performance in challenging cases and insufficient comparison and ablations. The rebuttal addressed all of them.

**Reviewer Concerns:**

Reviewer 3BJu requested detailed explanation of the pipeline, more experiments on NVIDIA dataset, more ablations and analysis for time consumption. Reviewer 1czs requested analysis of graph construction, more experiments on challenging cases and more ablations. Reviewer 57Ji requested comparison with 4DGS and complexity analysis. Reviewer SfVo requested experiments on challenging cases and analysis of computational cost. The concerns above were addressed by the rebuttal.

**Reviewer Scores:**

The paper initially received 6(3BJu), 4(1czs), 4(57Ji) and 6(SfVo). Apart from Reviewer 57Ji, the other three reviewers did not provide responses. Therefore, if all reviewers had participated in the discussion, I believe some ratings would have been raised. And I believe this paper has met the standards for acceptance.

---

### Decision · Program_Chairs · 2026-01-26

Accept (Poster)